

# Integrating multi-hazard susceptibility and building exposure: A case study for Quang Nam province, Vietnam

Chinh Luu[1*], Giuseppe Forino[2], Lynda Yorke[3], Hang Ha[4], Quynh Duy Bui[4], Hanh Hong Tran[5], Dinh Quoc Nguyen[6], Hieu Cong Duong[7], Matthieu Kervyn[8]

[1]Faculty of Hydraulic Engineering, Hanoi University of Civil Engineering, Hanoi, 100000, Vietnam
[2]School of Science, Engineering & Environment, University of Salford, Manchester, M5 4WT, UK
[3]School of Natural Sciences, Bangor University, Bangor, Gwynedd, LL57 2DG, UK
[4]Department of Geodesy, Hanoi University of Civil Engineering, Hanoi, 100000, Vietnam
[5]Faculty of Geomatics and Land Administration, Hanoi University of Minning and Geology, Hanoi, 100000, Vietnam
[6]Phenikaa University, Hanoi, 100000, Vietnam
[7]Hanoi University of Civil Engineering, Hanoi, 100000, Vietnam
[8]Department of Geography, Vrije Universiteit Brussel, Brussels, 1050, Belgium

*Correspondence to*: Chinh Luu (chinhltd@huce.edu.vn)

**Abstract.** Natural hazards have serious impacts worldwide on society, economy and environment. In Vietnam, throughout the years, natural hazards have caused a significant loss of lives as well as severe devastation to houses, crops, and transportation. This paper presents a new model for multi-hazard (floods and wildfires) exposure estimates using machine learning models, Google Earth Engine, and spatial analysis tools for a typical Quang Nam province, Vietnam case study. By establishing the context and collected data on climate hazards and impacts, a geospatial database was built for multiple hazard modelling, including an inventory of climate-related hazards (floods and wildfires), topography, geology, hydrology, climate features (temperature, wetness, wind), land use, and building data for exposure assessment. The hazard susceptibility and exposure matrices were presented to demonstrate a hazard profiling approach for multi-hazards. The results are explicitly illustrated for floods and wildfire hazards and the exposure of buildings. Susceptibility models using the random forest approach provide model accuracy of the AUC=0.882 and 0.884 for floods and wildfires, respectively. The flood and wildfire hazards are combined within a semi-quantitative matrix for assessing the building exposure to different combinations of hazards. Digital multi-hazard risk and exposure maps of floods and wildfires aid the identification of areas prone to climate-related hazards and the potential impacts of hazards. This approach can be used to inform communities and regulatory authorities on how they develop and implement long-term adaptation solutions.

## 1 Introduction

Different geographic areas worldwide, including mountainous, delta, and coastal regions, are facing distinct hazards and combinations of hazards (Rentschler et al. 2022). These challenges are intensified by population growth, urbanization, globalization, and climate change-induced shifts in extreme weather patterns, amplifying their adverse effects (Khatakho et al. 2021; Bangalore et al. 2018). Vietnam is becoming more vulnerable to natural hazards due to climate change and human



activities such as construction and urban development. While floods and storms represent the main hazards affecting Vietnam, risks for other hazards, such as landslides and wildfires, are also exacerbated by extreme climate patterns, land-use

change, and population expansion in Vietnam (Ipcc 2022). People who depend on natural resources lose their livelihoods and become more vulnerable (Balica et al. 2015).

Flooding is a common natural hazard in many coastal and low-lying areas wo   rldwide that can be caused by various factors, including heavy rainfall, snowmelt, storm surges, or the breaching of dams and levees (Viglione and Rogger 2015). In Vietnam, Quang Nam province is characterized by a coastal and low-lying topography and faces high flood risks due to

heavy rainfall, typhoons, and potential breaches of dams and levees (Luu et al. 2018). This issue holds particular significance for the Quang Nam province because flood events pose a direct threat to human lives and cause significant damage to its infrastructure, education, economic development, and health-related services (Lee et al. 2020).

Wildfires are also a natural hazard with devastating consequences, posing a severe threat to the environment and human communities (Tedim et al. 2015). Wildfires often occur due to a complex interplay of factors such as dry weather conditions,

high temperatures, low humidity, the presence of flammable vegetation, and other geo-environmental factors (Kalantar et al. 2020). Vietnam is particularly prone to fire events (Nguyen et al. 2023). According to the statistical data from the Global Forest Watch, Vietnam had a total of 674,612 forest fire alerts since 2012 and ranked sixth in Southeast Asia regarding forest fires in the last two decades (Ansori 2021).

The term "multi-hazard" is often closely linked to minimizing risks in most situations (Kappes et al. 2012). Multi-

hazards often interact in complex ways, and their combined impact might be greater than the sum of individual hazards (Wing et al. 2018). Broadening the assessment framework for these dynamic interactions can lead to a more comprehensive and accurate risk evaluation (De Angeli et al. 2022). Thus, multi-hazard susceptibility and exposure assessments are required for efficient disaster risk management (Zhou et al. 2015; Rusk et al. 2022). Multi-hazard susceptibility assessment provides insights into the spatial co-occurrence of multi-hazard (Rusk et al. 2022). Multi-hazard exposure assessment enables the

evaluation of the potential impact of multi-hazards on people, buildings, and critical facilities in a given location (De Angeli et al. 2022). This information is invaluable for emergency response planning, resource allocation, and the development of robust evacuation strategies (Kappes et al. 2012). Multi-hazard exposure assessment has become essential in building resilient societies, offering precise insights, optimizing responses, shaping policies, fostering adaptation, and empowering local communities to face the complexities of climate-related hazards (Abdullah and Sofyan 2023).

Advanced technologies, such as Machine Learning (ML), remote sensing, and big data analytics, play a critical role in predicting, monitoring, and mitigating the impact of increasing hazards (Velev and Zlateva 2023). ML algorithms can be applied to analyze historical data, climatic patterns, and geo-environmental characteristics to predict the likelihood and intensity of hazards (Kern et al. 2017). Satellite imagery aids in real-time monitoring of the Earth's surface changes, allowing for the timely detection of alterations in landscapes, weather patterns, and natural habitats and facilitating proactive

responses to potential hazards (Le Cozannet et al. 2020b; Parker et al. 2021). Relevant data analytics based on integrating diverse datasets offers a comprehensive view of various factors contributing to multi-hazards (Yu et al. 2018). This



integrated approach is crucial for understanding the interconnectedness of different variables and enhancing the precision of multi-hazard risk assessments (Ugliotti et al. 2023; Kwag et al. 2018). Technological approaches can be coupled with other human and social approaches to understanding the impacts of climate-related hazards and having appropriate responses and
adaptation (Lizarralde et al. 2021).

Google Earth Engine (GEE), a cloud-based geospatial processing platform developed by Google in 2010, offers an extensive and up-to-date archive of satellite imagery, robust analysis tools, custom ML algorithm development, and the capacity to integrate multiple data sources (Tamiminia et al. 2020). Classification And Regression Tree (CART) and Random Forest (RF) algorithms developed on the GEE platform are popular algorithms for both classification and regression
tasks (Kamal et al. 2019). In multi-hazard assessment, their flexibilities enable the integration of categorical and continuous variables into complex and nonlinear response models (Wang et al. 2022). Moreover, implementing these algorithms in the GEE environment can handle large-scale analyses and enhance processing speed and efficiency (Titti et al. 2022).

Various studies applied CART and RF algorithms in modelling natural hazard susceptibility and proved the success and accuracy in the performance (Chen et al. 2018; Kim et al. 2017). CART and RF have been used to build susceptibility maps
for single hazards, e.g., forest fire (Piao et al. 2022b) or landslide (Ilmy et al. 2021; Wu et al. 2022), but also in developing multi-geohazards (rock fall, landslide, and debris flow) susceptibility maps for Jiuzhaigou, China (Cao et al. 2020), or establishing the multi-hazard (forest fires and droughts) susceptibility maps for the Gangwon-do region in Korea (Piao et al. 2022a), or constructing the multi-hazard (flood, landslides, forest fire, and earthquake) susceptibility maps in Khuzestan Province, Iran (Pourghasemi et al. 2023). Most studies have focused on multi-hazard susceptibility assessment based on ML
models developed on the GEE platform, and they have indicated that these ML models exhibit good performance in estimating multi-hazard susceptibility but have not mentioned multi-hazard exposure assessment. Meanwhile, multi-hazard exposure assessment can help recognize overlapping exposures and comprehend the intricate relationships between several hazards (Wang et al. 2020).

Global South countries are more exposed to and affected by the impacts of disasters (Ibarrarán et al. 2009). Due to its
geographical location and unique natural conditions, Vietnam is exposed to various natural hazards: floods, landslides, droughts, and wildfires, further exacerbated by human activities combined with extreme weather conditions (Gan et al. 2021). Vietnam was ranked sixth among the countries most vulnerable to climate-related risks from 1999 to 2018. Additionally, it was ranked fifth among nations with the highest susceptibility to flood risks (Pham Quang and Tallam 2022). Predictably, climate change will increase the magnitude of natural hazards across many parts of Vietnam in the subsequent
time (Ipcc 2014). The central region of Vietnam, particularly Quang Nam province, is highly vulnerable to natural hazards, making sustainable development tasks very challenging (Giuliani et al. 2019). Floods intensified by tropical storms during the monsoon season (Luu et al. 2021; Ocha 2022) and wildfires exacerbated by dry seasons and high temperatures pose frequent threats and require comprehensive assessments of multi-hazard susceptibility and exposure in Quang Nam province (Duc Le and Thi Thu Vu 2013). Notwithstanding these longstanding issues with floods and wildfires in the Quang Nam





province of Vietnam, limited studies exist that evaluate both multi-hazard susceptibility and multi-hazard exposure assessment for the area.

Therefore, the study aims are (i) to present and apply a methodological approach to assess and map multi-hazards for the Quang Nam province; (ii) to utilize two ML models, CART and RF, that have been implemented on the GEE platform to build the multi-hazard (flood and wildfire) susceptibility maps for the Quang Nam province; (iii) to integrate the multi-
hazard susceptibility map with built environment data to assess the multi-hazard exposure; (iv) to assess the outputs as a multi-hazard/risks assessment tool and framework to aid risk reduction and management.

## 2 Study area

Quang Nam province is located in the central key economic region of Vietnam with significant economic growth and a huge potential for tourism. Since the "economic reforms" in 1986, Quang Nam province has seen significant socio-economic
transformations, such as the development of industrial zones and tourism. However, this fast development presents several issues for the province in pursuing sustainable development, necessitating optimal use of natural and socio-cultural resources (Chau et al. 2014).

Quang Nam province encompasses large topographic gradient types, from a coastal plain to steep mountains and various natural resources, with a total area of 10,438 km² (**Fig. 1**). Quang Nam had a total population of 1.84 million people in 2019,
with over 73% of the population residing in the coastal plain, which comprises just 25% of the total geographical area. The Kinh ethnic group comprises 92.3% of the population; the remainder consists of many ethnic minorities, including the Co Tu, Xo Dang, M'nong, Co, and Gie Trieng (Quang Nam Statistical Office 2019). Agriculture, forestry, and fisheries accounted for 56 % of the total labour force, although their contribution to the GDP is only 21.4% (Quang Nam Statistical Office 2019).
Quang Nam has a complex topography due to the Annamite Range extending to the sea, leading to strong separation in climate conditions, steep hills, and mountains. Terrain elevation tends to gradually lower from West to East, with mountainous areas (slope of $15^0$ or more) concentrated mainly in the West following the Annamite Range and the coastal plain running along the sea. The tropical monsoon climate is characterized by two distinct weather seasons in a year: the dry season from January to August, associated with water shortages, leading to droughts, and the rainy season from September to
December, often bringing excess water and leading to floods. Quang Nam has the highest annual rainfall in Vietnam. The average annual rainfall ranges from 2,200 mm to 2,700 mm, 70% falling during the rainy season (https://quangnam.gov.vn/. Types of natural hazards that often occur in Quang Nam province are floods, landslides, and wildfires (Duc Le and Thi Thu Vu 2013). Quang Nam province has recently been affected by wildfires and drought during the dry season. As a typical province in central Vietnam, Quang Nam represents many other provinces in Vietnam. This study, therefore, focuses on
assessing and mapping flood and wildfire hazards in Quang Nam province.



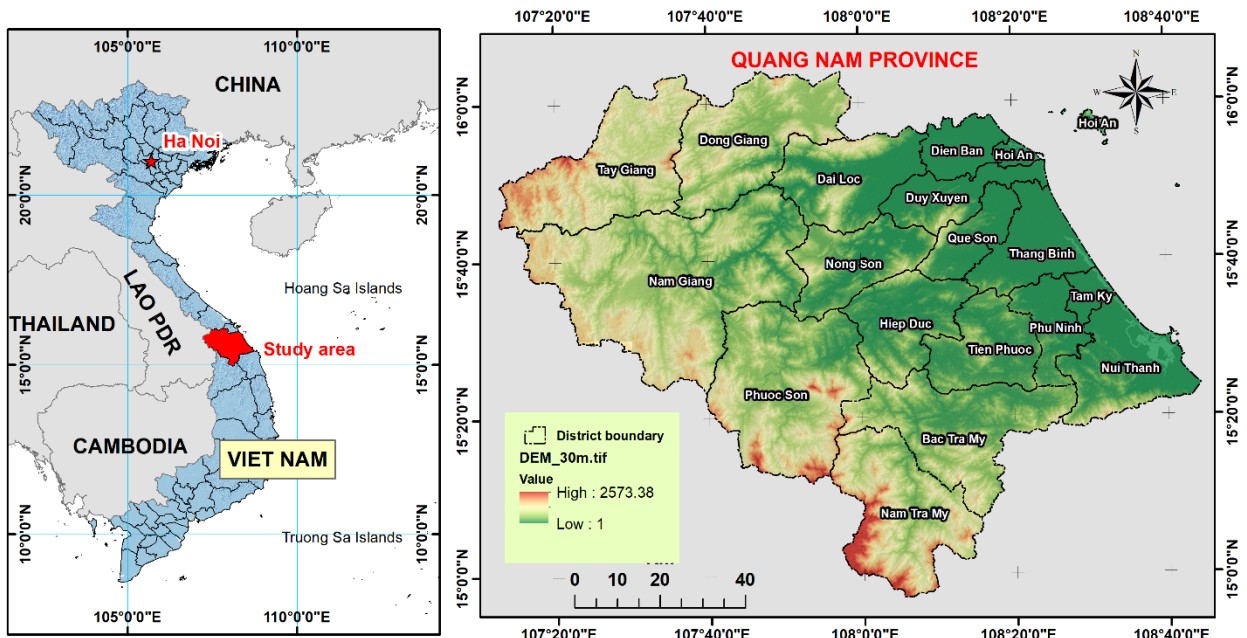

Figure 1. The study area, Quang Nam province in Vietnam (source: Shuttle Radar Topographic Mission Digital Elevation Model)

## 3 Methodology

### 3.1 Methodology flowchart

The implementation process comprises seven main stages, as follows: (1) Data collection for each hazard, (2) Building inventory maps of each hazard, (3) Checking multicollinearity of each input dataset, (4) Developing ML models to build susceptibility maps of each hazard, (5) Validating and comparing to select the best susceptibility map for each hazard, (6) Overlaying two best susceptibility maps to create a multi-hazard susceptibility map, and (7) Overlaying this multi-hazard susceptibility map with the building data to construct multi-hazard exposure map (**Fig. 2**). First, flood influencing and

wildfire influencing factors were collected, including topography, geology, hydrology, climate (temperature, wetness, wind), and land use based on their availability. Second, inventory maps of each hazard were created based on historical data collection. Third, the influencing factors of each hazard were tested for multicollinearity to enhance the reliability and stability of the model's predictions. Fourth, CART and RF models were developed on the GEE platform to construct susceptibility maps of each hazard. Fifth, the Area Under the ROC Curve (hereafter, AUC) was utilized to validate and

compare the predictive performance of the susceptibility map to choose the best model for each hazard. Sixth, the flood susceptibility map and the wildfire susceptibility map were combined to build a multi-hazard susceptibility map. Finally, this multi-hazard susceptibility map was overlaid with the building data to create a multi-hazard exposure map for the study area.



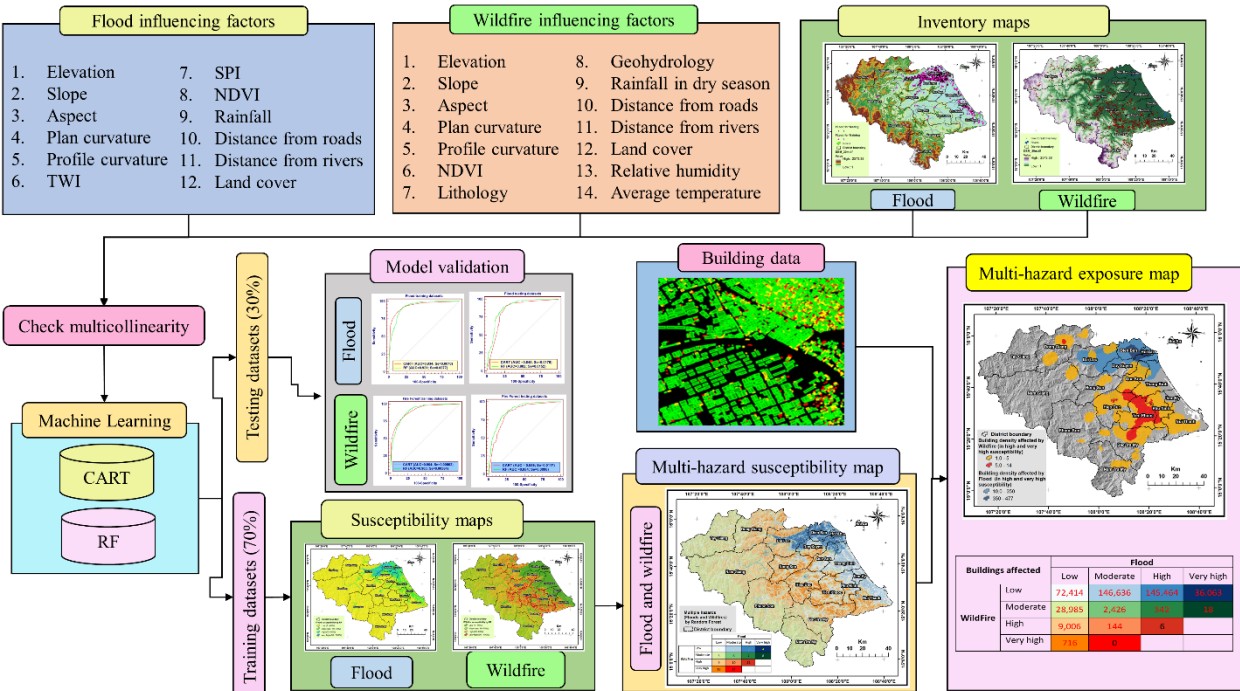

**Figure 2. Methodology flowchart for multi-hazard exposure mapping in this study.**

## 3.2 Data used

### 3.2.1 Inventories of floods and wildfires

The development of a reusable and accurate flood inventory is a crucial element in the susceptibility mapping of floods (Ahmadlou et al. 2018). The flood locations are collected from two sources. The first source is 847 historical flood marks of 2007, 2009, and 2013 historical flood events obtained from the Quang Nam Provincial Steering Committee of Natural Disaster Prevention and Control. A flood mark comprises a unique identifier, geographical coordinates (longitude and latitude), flood depth, and provider information. The second source is SAR data from Sentinel 1 for 2017 to 2021, which we compare with official reports from the Provincial Committee. We coded in Google Earth Engine to detect flood areas as in this study (Mai Sy et al. 2023). After that, the inundation areas of many years were overlayed and compared with the historical flood mark locations to avoid duplicates. Forty-seven new flood sites were detected and integrated with 847 historical flood marks for the inventory data.

In total, our flood inventory includes 871 flood locations: 70% of them (608 locations) were randomly selected to calibrate the flood susceptibility model, while the remaining 30% (263 locations) were designated for validating purposes (**Fig. 3**). In addition, a total of 871 non-flood locations were randomly selected across the study area using the "Create random point tool" in ArcGIS. The non-flood points were determined by overlaying all flood inventory onto the study area map. Non-flood points were only selected in zones that were outside all flood maps. Additionally, we implemented more



stringent criteria for the exclusion of non-hazardous points located on steep slopes (>10°) or areas of positive relief (such as hilltops) that are not susceptible to flooding. These non-flood points were then classified in a ratio of 70/30, mirroring the classification of the flood locations. This process was undertaken to create a comprehensive database for input into the GEE platform, which was utilized for modelling and validation purposes.

Satellite-based approaches are widely used to detect active fires and burnt regions (Le Cozannet et al. 2020a). This study involved the collection of 1,911 wildfire locations from various sources (**Fig. 3**). We collected the inventory data of fires from the online website of the National Forest Protection Department (available at https://watch.pcccr.vn/thongKe/diemChay). This agency utilizes data from many satellites (AQUA, J1, SUOMI, and TERRA) that are regularly received at the TeraScan receiving station located at the Forest Protection Department. Changes

in temperature and environment (humidity, wind direction) on the ground on the same day have been determined and duplicated at one fire location based on near-infrared bands of types of satellites to identify fire hotspots. We checked and filtered the duplicated wildfire locations, dates, and commune data field conditions for the filter. The obtained wildfire location data represents the specific fire sites captured by one type of satellite inside a particular commune at a given time. The area criterion is also important for obtaining accurate natural fire locations and eliminating human-made fire locations.

We used a filtration process to identify wildfire spots that exceed a minimum size threshold of 2 hectares based on the note field in the statistical data of the National Forest Protection Department. To determine the non-fire point, we assume that residential areas, water systems, and crop areas are locations that cannot burn. Based on these assumptions, we determine the corresponding non-fire points regarding quantity and distribution density by overlaying the wildfire inventory map onto the study area map.

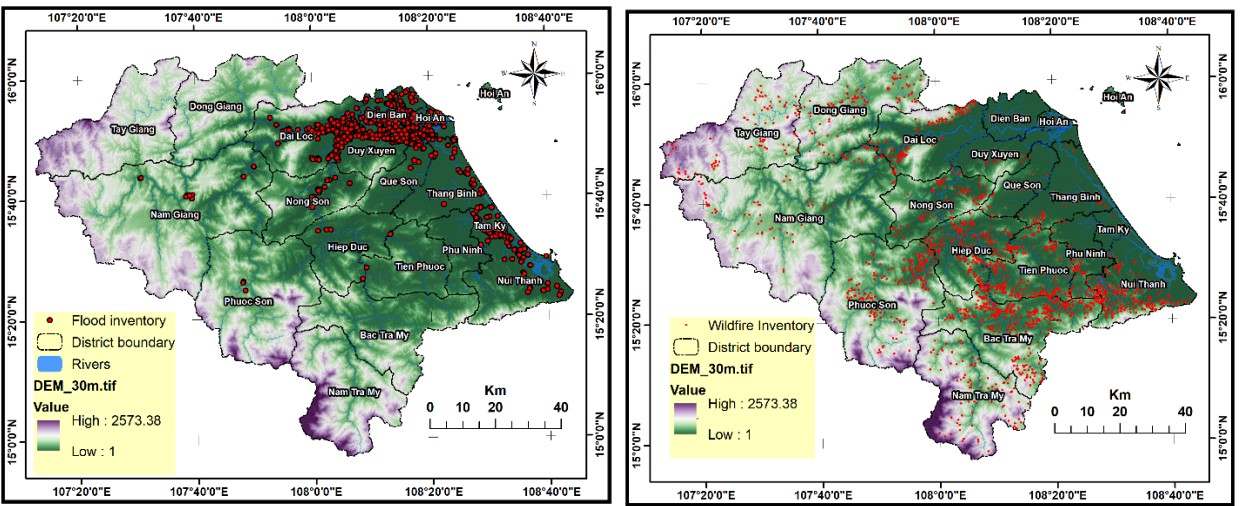


**Figure 3. Inventory maps of flood (left) and wildfire (right) in Quang Nam province.**



### 3.2.2 Influencing factors

**(i) Elevation**

Low-lying areas, often at lower elevations, act as natural drainage basins and are prone to flood occurrences as streams and
rivers flow from higher to lower elevations (Komolafe et al. 2020). High areas can act as natural barriers that slow the spread of fire events (Sibold et al. 2006). In this study, the elevation data was derived from the Shuttle Radar Topographic Mission (SRTM) Digital Elevation Model (DEM) with a 30-meter spatial resolution (https://earthexplorer.usgs.gov/). The altitude of the study area varies from 0 to 2,573 m (**Fig. 4a**).

**(ii) Slope**

Slope is another important terrain characteristic significantly influencing flood and wildfire occurrences (Pourghasemi et al. 2020). Steep slopes and increased flow velocity can lead to riverbank erosion and subsequent flooding downstream (Guo et al. 2022). In addition, flat slopes can lead to concentrated floods where water may stagnate or flow less rapidly (Zaharia et al. 2017). On the other hand, steep slopes can either mitigate or accelerate the spread of wildfires under the impact of wind (Eftekharian et al. 2019). The slope data is calculated from the DEM data (**Fig. 4b**).

**(iii) Slope aspect**

Slop aspect provides information about the direction of a slope face and may play a significant role in flood and fire formations (Vasilakos et al. 2009). In forestry, south-facing slopes in many areas are often drier and more susceptible to wildfires due to increased sunlight exposure and lower moisture levels (Adab et al. 2013). The aspect data is also calculated from the DEM (**Fig. 4c**).

**(iv) Slope curvature**

Slope curvature refers to the rate of slope change along the land's surface and contributes actively to flood and wildfire formations (Minár et al. 2020). Concave or depressional terrains (negative curvature) can trap water during heavy rainfall, leading to temporary ponding or small-scale flooding in these regions (Mohamed 2020). Concave curvature can accumulate dead plant material, creating a higher fuel load and increasing the hazard of forest fires (Banerjee 2021). This study
calculates the plan curvature and profile curvature from the DEM (**Fig. 4d and 4e**).

**(v) TWI and SPI**

The Topographic Wetness Index (TWI) is a topographic parameter used to quantify the propensity of accumulated water in a specific area (Meles et al. 2020). TWI helps identify low-lying areas in the landscape of accumulated water, making it valuable for flood hazard mapping (Nandi et al. 2016). Areas with higher TWI values generally indicate higher accumulated
water locations and higher moisture content in the soil and vegetation (Berhanu and Bisrat 2018). The TWI index can be calculated using the following equation (Beven and Kirkby 1979):

$$TWI = ln\left(\frac{A_S}{tan\,\varepsilon}\right) \tag{1}$$

where ε represents the slope angle in degrees and $A_S$ signifies the specific basin area in square meters per meter.



The Stream Power Index (SPI) quantifies the erosive power of flowing water in a stream network. SPI helps assess the potential for erosion and sediment transport within river channels (Zakerinejad and Maerker 2015). The SPI index is identified as follows (Moore et al. 1991):

$$SPI = A_S \times \tan \varepsilon \qquad (2)$$

This study calculates TWI and SPI indexes from the DEM (**Fig. 4f** and **Fig. 4g**).

**(vi) NDVI**

The Normalized Difference Vegetation Index (NDVI) assesses the density of vegetation by calculating the disparity between near-infrared and red light wavelengths (Bhandari et al. 2012). The sudden decrease in NDVI values could signify changes in vegetation health due to natural hazards (flood, fire, drought, landslide) or human activities (Teodoro and Duarte 2022). Areas with high NDVI values may indicate dense vegetation, especially during dry seasons, which can act as significant material for forest fires (Lambert et al. 2015). The NDVI can be calculated as follows:

$$NDVI = \frac{NIR - Red}{NIR + Red} \qquad (3)$$

where *NIR* denotes the near-infrared segment of the electromagnetic spectrum (750–1100 nm), and *Red* corresponds to the red segment of the electromagnetic spectrum (600–750 nm).

This study calculated the NDVI index from the Landsat 8 imagery (**Fig. 4h**).

**(vii) Distance from roads and distance from rivers**

Distance from roads is a potential controlling factor in flood and wildfire occurrences. They can exacerbate both flood and forest fire events because roads may serve as pathways for water runoff during heavy rainfall and for fire spread in dry conditions (Yousefi et al. 2020). Distance from roads and associated embankments can hamper natural floodplains, causing accumulated water during heavy rains (Douven and Buurman 2013). Moreover, roads with impermeable surfaces can increase surface runoff by preventing water from infiltrating the ground (Yu et al. 2021). Distance from roads can generate potential ignition sources and facilitate firefighting movement (Wang et al. 2015). The 1:50,000 study area road network map was created from the 2019 national road network map from the Department of Survey, Mapping, and Geographic Information (**Fig. 4i**).

Distance from rivers may influence flood and fire occurrences due to their dynamic relationships with topography, hydrology, and vegetation (Pouyan et al. 2021). Rivers naturally overflow during heavy rainfall, making neighbouring areas and floodplains highly susceptible to flooding (Desalegn and Mulu 2021). Rivers attract human settlements and recreational activities (Gibeau et al. 2002), so areas near rivers are prone to ignition from human-induced sources, especially during dry seasons (Ye et al. 2017). However, distance from rivers also ensures a readily available water supply and makes ground/vegetation wet due to shallow groundwater, reducing wildfire susceptibility. The study area's river network map in the 1:50,000 scale was also collected from the Department of Survey, Mapping, and Geographic Information in 2019 (**Fig. 4k**).



**(viii) Land cover**

Different land cover types have varying abilities to absorb water, so they may also contribute to the occurrence of floods and

wildfires (Agus et al. 2020). Natural land cover features such as floodplains and wetlands act as natural buffers during floods (Fasching et al. 2019). Loss of these areas due to urbanization or deforestation increases the occurrence frequency of flood events (Cirella and Iyalomhe 2018). Different land cover types, such as dense forests, grasslands, shrublands, and dead vegetation, contribute to accumulating materials for fires (Agus et al. 2020). This study extracted the land cover data from Sentinel-2 optical imagery for 2021 using the deep learning method (https://livingatlas.arcgis.com/landcover/) (**Fig. 4j**).

**(ix) Lithology and geohydrology**

Lithology studies bedrock types and their mineralogical properties, significantly influencing soil composition (Gray et al. 2016). The lithological characteristics of an area can indirectly influence wildfire behaviour (Pourghasemi et al. 2020). Some rock types, such as shale or coal, can affect the spreading rate of wildfire events (Lu et al. 2021). Lithology also affects the permeability of geological formations and directly contributes to flood occurrences (Jansen 2006). Impermeable rocks, like

crystalline rock or bedrock, can facilitate increased surface runoff during heavy rainfall, resulting in the formation of floods or flash floods (Langston and Temme 2019). The 1:50,000 lithological map of Quang Nam province was provided by the Ministry of Natural Resources and Environment of Vietnam in 2021, including nine classes: Magma Neutral Intrusive Rocks (XNA), Alumosilicate Metamorphic Rocks (BCA), Detrital Sedimentary Rocks (DTU), Quartz-Rich Greenstone Metamorphic Rocks (TTA), Ultramafic Volcanic Eruption Rocks (PTM), Carbonate Rocks (CAR), Mafic-Ultramafic

Intrusive Rocks (XNM), Neutral Volcanic Eruption Rocks (PTA), Quartz-Rich Metamorphic and Volcanicclastic Rocks (BTT) (**Fig. 4l**).

Geohydrology studies the movement and availability of groundwater and plays a vital role in influencing vegetation development (Orellana et al. 2012). Geohydrology plays a crucial role in understanding and predicting flood formations based on the presence of aquifers (Lauber et al. 2014). The movement and distribution of groundwater directly impact the

behaviour of surface water during heavy rainfall (Chen and Hu 2004). In addition, geohydrology indirectly influences forest fire occurrences because of its impact on soil moisture, land subsidence, and aquifer characteristics (Wösten et al. 2008). Low groundwater levels due to geological formations may lead to dead vegetation, leading to higher susceptibility to ignition and fire spread (Hasan et al. 2023). The geohydrological map at a 1:50,000 scale was provided by the Ministry of Natural Resources and Environment of Vietnam in 2020 (**Fig. 4 (q)**).

**(x) Rainfall**

In both flood and wildfire occurrences, the amount, intensity, and duration of rainfall may play a role (Stoof et al. 2012). Heavy and prolonged rainfall can lead to increased water flow into rivers and streams and can contribute to the complex dynamics of flood distribution (Khan 2013). In contrast, insufficient rainfall over an extended period leads to drought conditions, drying out forests and creating ideal conditions for wildfires (Cochrane and Barber 2009). The rainfall data was

recorded from 2002 to 2022 and collected from 33 rain gauge stations in Quang Nam province. This study used the inverse distance weighted technique to generate rainfall maps for the rainy and dry seasons separately (**Fig. 4m** and **Fig. 4n**).



### (xi) Temperature

The average monthly temperature in the dry months is often closely related to wildfire occurrences (Kumari and Pandey 2020). In a climate change context, higher average temperatures can increase evaporation and transpiration rates, drying out vegetation that can facilitate fires to ignite and spread rapidly (Houston 2006). Rising high temperatures can extend the duration of fire events (Sun et al. 2019). The temperature data were collected from March to August between 2020 and 2023 (dry seasons) at https://power.larc.nasa.gov/data-access-viewer/. This research used the inverse distance weighted approach to produce a temperature map specifically for dry seasons (**Fig. 4o**).

**Table 1 Potential factors affecting flood and forest fire in Quang Nam province (where X indicates an influencing factor, 0 indicates no influence).**

| No. | Used factors | Flood influencing factors | Wildfire influencing factors |
|-----|--------------|---------------------------|------------------------------|
| 1 | Elevation | X | X |
| 2 | Slope | X | X |
| 3 | Aspect | X | X |
| 4 | Plan curvature | X | X |
| 5 | Profile curvature | X | X |
| 6 | TWI | X | O |
| 7 | SPI | X | O |
| 8 | NDVI | X | X |
| 9 | Distance from roads | X | X |
| 10 | Distance from rivers | X | X |
| 11 | Land cover | X | X |
| 12 | Average rainfall | | |
|    | - In rainy season | X | O |
|    | - In dry season | O | X |
| 13 | Average temperature | O | X |
| 14 | Lithology | X | X |
| 15 | Geohydrology | X | X |















**Figure 4. Influencing factors to flood and wildfire: (a) Elevation, (b) Slope, (c) Aspect, (d) Plan curvature, (e) Profile curvature, (f) TWI, (g) SPI, (h) NDVI, (i) Distance from roads, (j) Land cover, (k) Distance from rivers, (l) Lithology, (m) Rainfall, (n) Rainfall in dry season, (o) Temperature, and (p) Geohydrology.**

### 3.2.3 Built environment data

In this study, we use the building data to assess the potential impact of natural hazards such as floods and wildfires on building infrastructure, considering housing/building a key livelihood asset. Spatial data on the building infrastructure of Quang Nam province is extracted from the Open Building dataset of Google (https://developers.google.com/earth-engine/datasets/catalog/GOOGLE_Research_open-buildings_v3_polygons). The collection contains information about each building, including a polygon representation of its footprint on the ground and a confidence score showing the level of certainty about its classification as a building (Sirko et al. 2021). We filtered the data with a confidence level of more than





80% and an area larger than 30m for accurate data on buildings. The data was checked with Google Earth, and it is well-integrated. The data is created from high-resolution satellite photography with a resolution of 50 centimetres.

### 3.3 Methods used

**3.3.1 Machine learning approach for hazard susceptibility modelling**

In this study, two ML models, including CART and RF, have been developed on the GEE workspace to construct the multi-hazard (flood and wildfire) susceptibility maps for the Quang Nam province.

The CART was first introduced by Breiman et al. (1984). It is an algorithm used for both classification and regression tasks (Johns et al. 2021). CART builds binary trees recursively by splitting the dataset into subsets based on the feature

values (Tang and Zhang 2020). Mathematically, this algorithm can be summarized as follows (Ahmadlou et al. 2022):

1. Input a training dataset $D = (X, Y)$ where $X$ is the feature variable and $Y$ is the target variable (class labels for classification, numerical values for regression).

2. For the classification issue, the CART algorithm uses the Gini impurity coefficient on these subsets to measure the disorder or impurity of an input dataset. The Gini impurity coefficient is determined using the following equation:

$$Gini(D) = 1 - \sum_{i=1}^{N} P_i \qquad (4)$$

where $Gini(D)$ is the Gini impurity coefficient of the input dataset $D$, $N$ represents the number of classes in the input dataset, and $P_i$ denotes the probability of class $i$ in dataset $D$.

The CART continues seeking the best feature and threshold recursively until a stopping criterion, such as maximum tree depth (*max_depth*) or minimum samples in a leaf (*min_samples_leaf*). After that, the resulting tree can be used to classify new.

Like all decision tree algorithms, CART is prone to overfitting, especially when the tree becomes too deep (Rasekhschaffe and Jones 2019). To mitigate this, pruning techniques and hyperparameter tuning are often applied to optimize the tree's structure, ensuring generalizability to unseen data (Ahmadlou et al. 2022). A combination of pruning techniques and hyperparameter tuning of the CART algorithm was implemented. The pruning technique involves removing certain branches of the tree that do not contribute significantly to predictive accuracy, thus ensuring the model suits the

training dataset. Hyperparameter tuning was employed to explore different tree depths to identify the optimal depth of the tree. This required varying the maximum depth parameter while assessing the model's performance on a testing dataset.

The RF is a widely-used ML algorithm developed by Breiman (2001), which combines the output of multiple decision trees to reach a single result (Naghibi et al. 2016). It is used for both classification and regression tasks (Genuer et al. 2010). The content of this technique can be described as follows (Breiman 2001):

1. Input a training dataset $D$ of $N$ bootstrap samples, $D = (X, Y)$ where $X$ is the feature variable and $Y$ is the target variable (class labels for classification, numerical values for regression). The RF technique creates multiple decision trees



using bootstrapped subsets of the training data *D*. Each tree is constructed using *N* samples drawn with replacement (bootstrap sampling).

2. For each tree and at each split, a subset of features (*m*) is randomly selected from the total number of features in the training dataset (*M*) to ensure diversity among the trees.

3. Each tree in the RF algorithm is built using the selected bootstrap sample and features in the first and second steps. The tree is developed by recursively dividing the dataset based on the selected features and splitting criteria.

4. The RF technique combines these predictions (multiple decision trees) due to the specific tasks. The mode of prediction from individual trees is taken as the final prediction for classification tasks.

Generally, the diversity introduced by randomness in feature selection and bootstrapped samples helps this algorithm to generalize well to unseen data (Kanevski et al. 2009), reducing overfitting and improving the overall model performance (Feng et al. 2020; Zermane and Drardja 2022). In this study, a combination of random feature selection and bootstrapping of the RF algorithm was conducted. The random feature selection technique was used for feature selection by considering a subset of features at each split during the construction of the decision tree. This ensures the model is exposed to different features, preventing it from becoming overly dependent on a specific subset. The bootstrapping technique was applied to create diverse training datasets for each tree in the RF.

### 3.3.2 An experimental process on the GEE platform

The GEE (https://earthengine.google.com/) is a valuable cloud-based platform for assessing and analyzing climate-related hazards because of its scalability and analytical flexibility (Nakalembe et al. 2021). The GEE cloud computing platform was employed for the pixel-based CART and RF algorithms to build susceptibility maps for each hazard, including floods and wildfires. The input data was collected from various sources and formats. First, we pre-processed and converted these data into raster format with the 30-meter spatial resolution in the GIS environment. Then, these data were uploaded into the GEE platform. The conduct steps were represented as follows:

1. All thematic maps were uploaded, overlaid, and gathered into each dataset for modelling flood and wildfire susceptibility.

2. Inventory maps of floods and wildfires were uploaded and connected with the corresponding attributes.

3. These inventory maps were classified into two datasets: the training dataset with 70% inventory locations and the testing dataset with 30% remaining inventory locations.

4. The CART and RF models were developed to construct each hazard susceptibility map on the training dataset. The RF technique was also employed to check the importance of input variables.

5. The ROC curve and AUC value were applied to validate and compare the predictive performance of each hazard susceptibility model on the testing dataset to select the best predictive model for each hazard.

6. The most accurate susceptibility maps of each hazard were combined to create a multi-hazard susceptibility map.



7. Finally, this multi-hazard susceptibility map was overlaid with the building data to generate a multi-hazard exposure map for the research area.

### 3.3.3 multicollinearity and variable importance

Variance Inflation Factors (VIF) and Tolerance are critical statistical measures in detecting the presence of multicollinearity among input variables (Arabameri et al. 2018). VIF quantifies how much the variance of an estimated regression coefficient increases due to multicollinearity (Ma et al. 2020). Tolerance is the reciprocal of VIF and reflects the proportion of variance in a predictor that is not forecast from other predictors (Bui et al. 2023). Significant multicollinearity among input variables is detected if the VIF value surpasses 10 or the Tolerance value drops below 0.1 (Miao et al. 2023).

### 3.3.4 Model validation and comparison

This study used the ROC curve and AUC to validate the predictive performance of each hazard susceptibility model, including CART and RF models. The ROC curve is generated by plotting the true positive rate (sensitivity) against the false positive rate (1-specificity) for different threshold values (Carter et al. 2016). Sensitivity quantifies the ability of the model to correctly identify susceptible areas, while specificity measures the capability to identify non-susceptible areas (Meghanadh et al. 2022) correctly. The AUC is calculated to quantify the quality of the predictive model (Carter et al. 2016). The AUC values vary from 0 to 1, where AUC values of 0.5–0.6 reflect a low predictive performance, 0.7-0.8 is interpreted as a medium predictive performance, 0.8-0.9 indicates good predictive performance, and 0.9-1.0 denotes excellent predictive performance (Wang et al. 2016).

## 4 Results

### 4.1 Assessment of multicollinearity and variable importance

In this research, the VIF and tolerance values of influencing factors to flood and wildfire susceptibility modelling are satisfactory, so all input factors are selected to develop multi-hazard susceptibility maps (**Table 2**). In natural hazard susceptibility modelling, each input variable may influence the occurrences of each hazard in various ways (Pourghasemi et al. 2020). Variable importance assessment can identify which factors have the most significant impact on the multi-hazard formations (Javidan et al. 2021). Moreover, this operation can enhance the precision of multi-hazard prediction models, making them more reliable for risk assessments (Pourghasemi et al. 2020). RF is one of the most popular ML algorithms for evaluating variable importance by measuring how much they contribute to the model's accuracy (Fox et al. 2017). Thus, this technique was applied to assess the importance of all input variables. The obtained results in **Table 2** show that rainfall (weight = 0.1742), distance from rivers (weight = 0.1620), NDVI (weight = 0.1330), and land cover (weight = 0.1159) are the indicators that significantly contribute to control the spatial distribution of flood events within the study area.

**Table 2 Assessment of multicollinearity and variable importance to flood influencing factors.**



| Factors | Flood | | | |
|---|---|---|---|---|
| | Tolerance | VIF | Variable importance | Rank |
| Rainfall | 0.832 | 1.225 | 0.1742 | 1 |
| Distance from rivers | 0.945 | 1.204 | 0.1620 | 2 |
| NDVI | 0.759 | 1.774 | 0.1330 | 3 |
| LULC | 0.582 | 2.160 | 0.1159 | 4 |
| Aspect | 0.98 | 1.019 | 0.1095 | 5 |
| TWI | 0.725 | 1.676 | 0.0753 | 6 |
| Distance from roads | 0.600 | 3.241 | 0.0709 | 7 |
| Plan Curvature | 0.798 | 3.669 | 0.0695 | 8 |
| Profile Curvature | 0.876 | 1.418 | 0.0320 | 9 |
| Elevation | 0.777 | 1.259 | 0.0300 | 10 |
| Slope | 0.748 | 2.106 | 0.0270 | 11 |
| SPI | 0.948 | 1.117 | 0.0007 | 12 |

395    The received results presented in **Table 3** demonstrate that temperature (weight = 0.1784), distance from rivers (weight = 0.1112), NDVI (weight = 0.1089), and distance from roads (weight = 0.1065) are the parameters that have a significant impact on the formation of wildfire events within the study area.

**Table 3 Assessment of multicollinearity and variable importance to wildfire influencing factors.**

| Factors | Wildfire | | | |
|---|---|---|---|---|
| | Tolerance | VIF | Variable importance | Rank |
| Temperature | 0.643 | 1.555 | 0.1784 | 1 |
| Distance from rivers | 0.697 | 1.435 | 0.1112 | 2 |
| NDVI | 0.835 | 1.198 | 0.1089 | 3 |
| Distance from roads | 0.472 | 2.118 | 0.1065 | 4 |
| Slope | 0.512 | 1.954 | 0.0953 | 5 |
| Rainfall in dry season | 0.384 | 2.603 | 0.0739 | 6 |
| LULC | 0.737 | 1.356 | 0.0613 | 7 |
| Profile Curvature | 0.786 | 1.273 | 0.0538 | 8 |
| Elevation | 0.524 | 1.909 | 0.0500 | 9 |
| Plan Curvature | 0.715 | 1.398 | 0.0481 | 10 |
| Aspect | 0.513 | 1.948 | 0.0473 | 11 |
| Lithology | 0.551 | 1.816 | 0.0420 | 12 |
| GeoHydrology | 0.636 | 1.572 | 0.0233 | 13 |



## 4.2 Flood susceptibility map and model validation

For flood susceptibility models, the ROC curve analysis on the training dataset signifies that the CART model has the highest value of AUC (0.934), and the RF model has a lower AUC (0.921). The ROC curve analysis on the validation dataset reveals that the AUC value of the RF model (0.882) is higher than that of the CART model (0.845). This result demonstrates that the RF model has the best predictive performance for flood susceptibility mapping (**Fig. 5**).

Since the RF shows good predictive performance, it is selected to generate the flood susceptibility map for the research area with the training dataset. The flood susceptibility map delineates the different geographical zones with increasing levels of susceptibility to flood events. We use the quantile method for classifying the susceptibility values with low (0-40%), moderate (40-70), high (70-90%), and very high (90-100%) and set the green-blue colour scheme for flood susceptibility (**Fig. 6**). The high and very high susceptibility areas are along the river basins consistently with the flood inventory shown in **Fig. 3**.

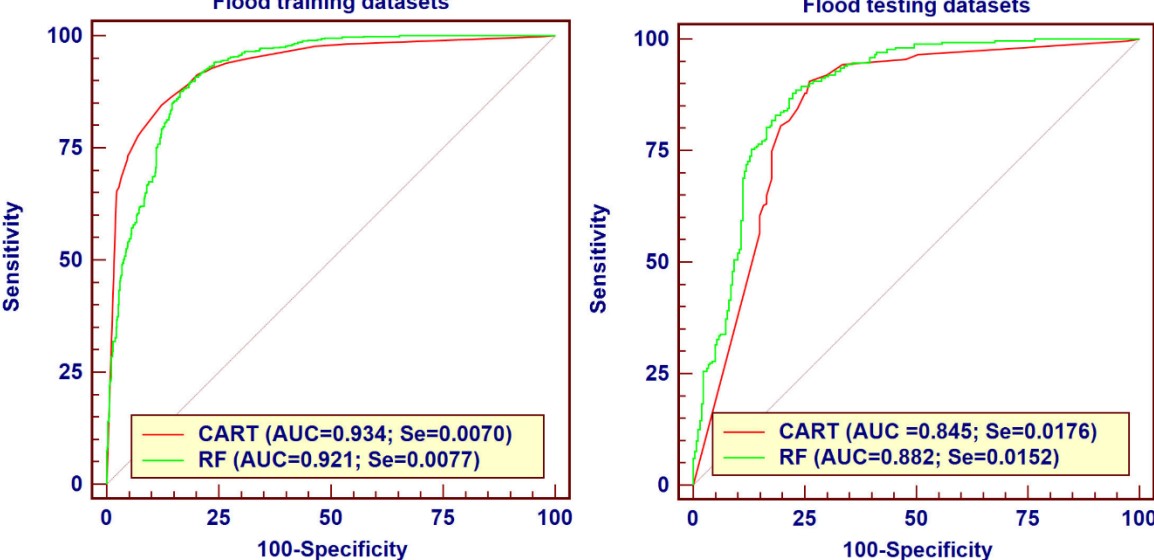

**Figure 5.** ROC curve and AUC analysis result from flooding susceptibility models.





**Figure 6.** The flood susceptibility map was derived using the RF model for Quang Nam province.

## 4.3 Wildfire susceptibility map and model validation

The ROC curve analysis on the training dataset for wildfire susceptibility models denotes that both the CART and RF models have the same AUC value (0.905). In contrast, the ROC curve analysis on the validation dataset reveals that the AUC value of the CART model (0.846) is lower than that of the RF model (0.884). This result reflects that the RF model is the best forecast model for wildfire susceptibility mapping (**Fig. 7**).

Given the satisfactory predictive performance shown by the RF model, it has been chosen as the preferred method for generating fire susceptibility maps for the study area using the provided training dataset. The wildfire susceptibility map demarcates the diverse levels of susceptibility to fire occurrences. The same quantile approach is used to categorize susceptibility values. A green-yellow-red color scheme represents wildfire susceptibility (**Fig. 8**). The high-prone areas of wildfire hazard are in the middle high-land, not in the very high mountainous nor the lowland areas.



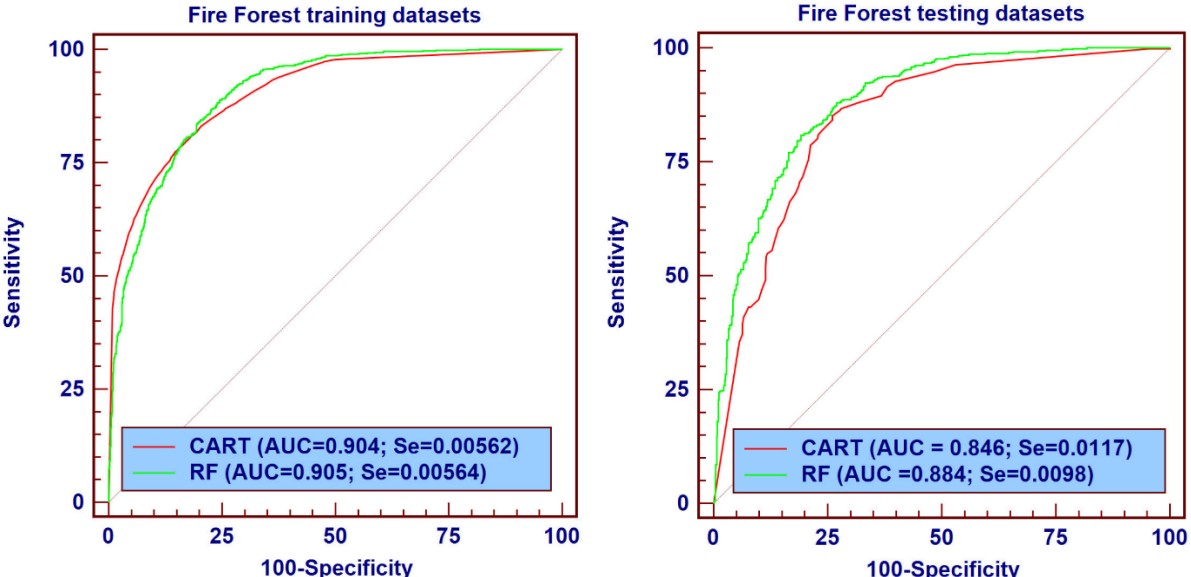

**Figure 7.** ROC curve and AUC analysis result from wildfire susceptibility models.

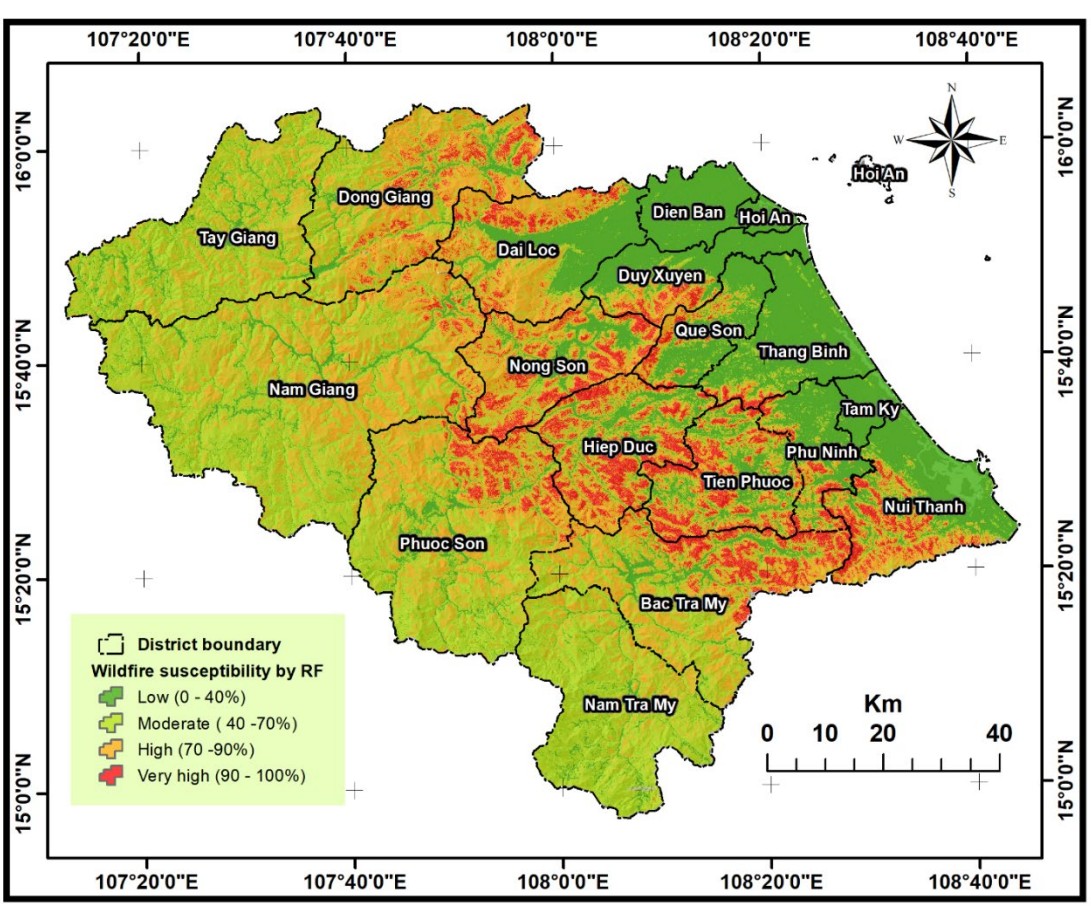



**Figure 8.** The wildfire susceptibility map was derived using the RF model for Quang Nam province.

## 4.4 Multi-hazard susceptibility and exposure mapping

430    The multi-hazard susceptibility map for Quang Nam province was generated by examining the interplay between wildfire and floods. The map depicts a matrix-based classification which enables the definition of new susceptibility classes (low, moderate, high, very high) of combined hazards and provides a unique multi-hazard profile for each location (**Fig. 9**). In the matrix, not all combinations of hazards are existing, as there is no area with high susceptibility of floods and high susceptibility of wildfires. Combining the multi-hazards through a matrix gives a good visual for an overview of multi-

435    hazards for the large scale of the whole province. The multi-hazard susceptibility map shows that the areas with very high wildfire susceptibility have low flood susceptibility and *vice versa*. The lowland coastal area is characterized by moderate to very high flood hazards but limited fire hazards (categories 2, 3, 4). The mid-altitude slopes are categorized by low flood but high to very high fire hazards (categories 9-10), with the exception of possible floods along the main valleys, and the upland is associated with moderate to low levels of the two hazards (categories 1-5).

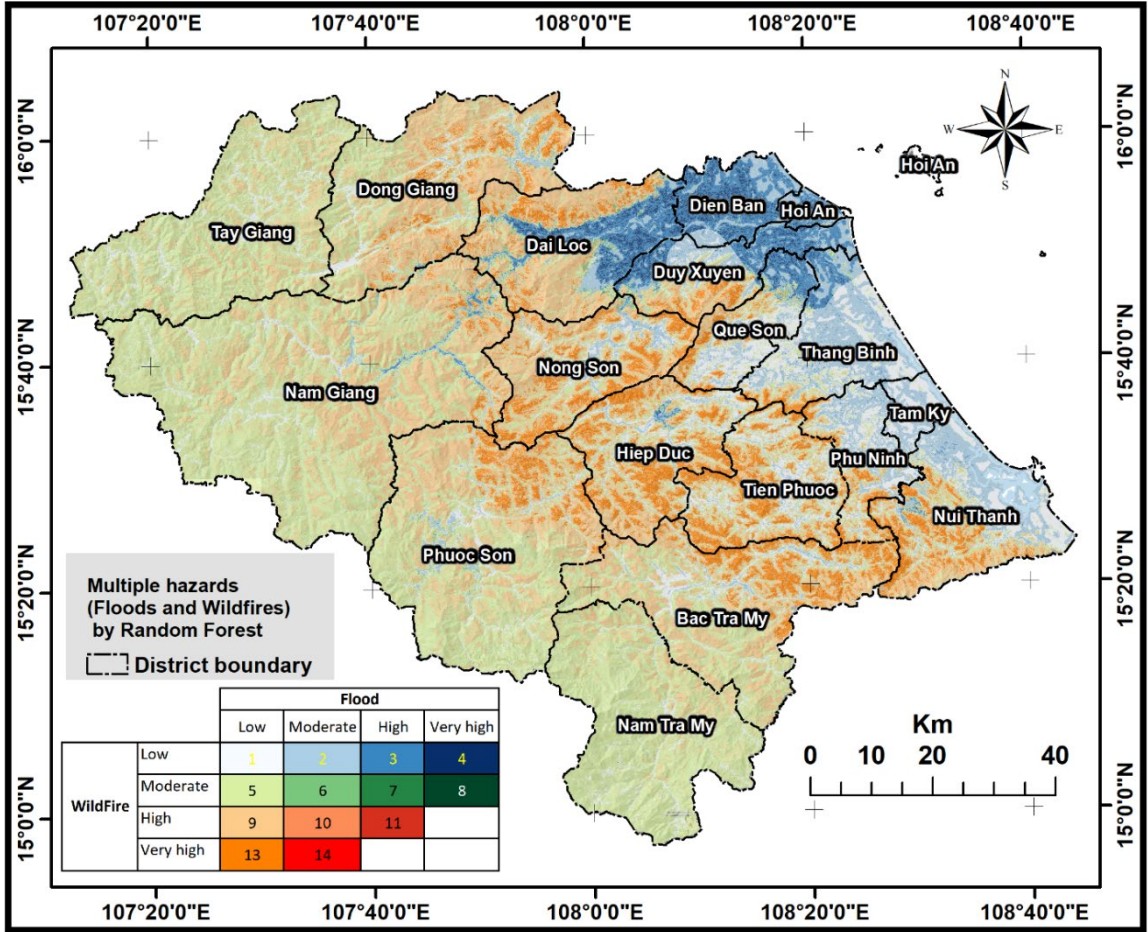

440



**Figure 9.** Integrated multi-hazard susceptibility classification combining flood and wildfire using random forest for Quang Nam province.

Our analysis examines the optimal sequence for integrating the two hazards, followed by assessing the exposure in conjunction with buildings. The matrix of the number of buildings and area affected by each hazard level is converted into the percentage of total buildings in each cell of multi-hazard levels. We can compare the two in **Table 4**. It is highlighted that the proportion of buildings in very high flood - low fire susceptibility category is much larger than the area of this category. In contrast, the proportion of buildings in category 13 (low flood - very high fire susceptibility) is much smaller than the area fraction. This highlights that measures to limit the impact on buildings (and so on people) to limit flood are much more important than for fire.

**Table 4 Statistics of the percentage of buildings affected and the percentage of area represented in each cell by flood and wildfire hazard in Quang Nam province.**

| Buildings affected (%) | | Flood | | | |
|---|---|---|---|---|---|
| | | Low | Moderate | High | Very high |
| **WildFire** | Low | 16.375 | 33.159 | 32.894 | 8.155 |
| | Moderate | 6.554 | 0.549 | 0.077 | 0.004 |
| | High | 2.037 | 0.033 | 0.001 | |
| | Very high | 0.162 | 0 | | |

| Area affected (%) | | Flood | | | |
|---|---|---|---|---|---|
| | | Low | Moderate | High | Very high |
| **WildFire** | Low | 9.605 | 8.353 | 3.920 | 1.010 |
| | Moderate | 38.587 | 0.559 | 0.021 | 0.000 |
| | High | 29.966 | 0.118 | 0.000 | |
| | Very high | 7.859 | 0.002 | | |

## 5 Discussion

Multi-hazard susceptibility and exposure assessments are crucial and multifaceted in disaster management and community resilience (Menoni et al. 2012). In this study, floods and wildfires are used as examples of two hazards with different spatial patterns but quite similar spatial extent and frequency to highlight that they have a very different impact on build-up infrastructure. Additional hazards, such as landslides or droughts, should be added to the scheme, with little to multi-dimension hazard matrix/profiling of each zone. This would help define the hazard profile for each zone and identify which areas are indeed affected by multiple and maybe combining hazards (Yousefi et al. 2020; Nachappa et al. 2020).

ML models have been extensively used in diverse hazard evaluations, such as floods, landslides, and wildfire susceptibility (Bui et al. 2022; Ha et al. 2022; Pourtaghi et al. 2016). These techniques are advantageous in evaluating the efficacy of different models under comparable circumstances, considering similar influencing elements. This approach



ensures a fair and unbiased determination of the most appropriate model for addressing a specific danger within a particular location. The modelling and mapping of multi-hazard susceptibility often rely on a system of multifaceted and multi-scaled natural factors, encompassing topography, geo-hydrology, environment, and hydro-meteorology conditions within the
research area (Tavakkoli Piralilou et al. 2022).

Our research analyzed the combined exposure to flood and wildfire hazards in Quang Nam province, Vietnam. Utilizing ML models (CART and RF) to assess the multi-hazard susceptibility, we can show that the RF model exhibited comparable levels of accuracy for both flood and wildfire hazards. Additionally, both models demonstrated good performance for flood and wildfire susceptibility maps, aligning with earlier research findings (Hasanzadeh Nafari et al. 2016; Nachappa et al.
2020). The accuracy of a model is dependent on the selection of the influencing elements used in mapping natural hazard susceptibility (Pourtaghi et al. 2016). This study carefully checked multicollinearity for influential factors and variable importance was measured to find the most suitable factors for the modelling input. In addition, the selection of the non-hazard points is also thoroughly carried out with the specific standards, contributing to better modelling performance.

The integration of the susceptibility maps of flood and wildfire hazards into a multi-hazards susceptibility matrix
highlights that flood and wildfire events threaten different areas and proportions of the entire Quang Nam province. The multi-hazard map is built upon a susceptibility class matrix for flood and wildfire events instead of a simple summation of both susceptibility maps. Indeed, the matrix enables the identification of regions with different combinations of hazard susceptibility for floods and wildfires. The exposure maps generated by combining the susceptibility map with the built environment data exhibit the total affected housing for different susceptibility levels of each hazard and muti-hazards.
Creating a multi-hazard exposure map that effectively delineates regions susceptible to floods and landslides via the implementation of a matrix-based approach and combining the map with built environment data to assess the exposure elements of the hazards has not previously been attempted by other researchers has not addressed the exposure of multi-hazard yet, only the hazard aspects. The combination with exposure highlights that different districts have to deal with different combinations of hazard susceptibility and that exposure to fire is much lower than flood hazards despite the broad
spatial distribution of the wildfire susceptibility.

Our findings suggest that ML models such as CART and RF should be used to analyze multi-hazard exposure for various geographical areas particularly susceptible to recurring incidents of wildfire and floods. Our data has shown these tools to model risk and exposure effectively. The evaluation of the availability of influencing variables, as well as their fit in multi-hazard exposure assessment, is an intriguing aspect to consider. The multi-hazard exposure maps for Quang Nam
province offer valuable insights to planners, disaster management specialists, and regional authorities, enabling them to adopt more effective management strategies for minimizing the many hazards present in the area. This approach may also facilitate the development of comprehensive strategies that address areas of high vulnerability to both hazards rather than focusing on individual hazards.



## 6 Conclusion

The study produced a novel and integrated approach to assessing the climate hazards of floods and wildfires. We explored multi-hazards assessment and risk through an ML and modelling approach. Through investigation of the magnitude of flood and wildfire hazards and the impacts of those hazards on the built environment, our modelling approach consisted of collating a database of topography, climate, geology and environment data to input into our model and developing advanced ML models for multi-hazards modelling and coding in GEE to produce credible susceptibility and risk maps. The

susceptibility evaluation incorporated a matrix that combined hazards associated with flooding and wildfires. Through the integration of built environment data with our multi-hazard map, this facilitated an assessment of the potential exposure to multi-hazards across the region. Going forward, the potential for digitally-generated, multi-hazard and exposure maps for other climate-related hazards, such as landslides or drought, would further aid the identification of regions susceptible to these disasters and facilitate a rapid assessment of the consequences resulting from these events. This research has

demonstrated that effective maps can be developed using readily-available and accessible data and ML tools that should help inform both communities and regulatory authorities in Vietnam and beyond about the likelihood of risk and impacts from climate-related hazards. This research has the potential to provide clear information that will inform the development and implementation of long-term risk reduction and adaptation strategies.

### Acknowledgements

We sincerely thank the British Council for funding the UK/Viet Nam Season 2023 project "Evaluation of risks from multi-natural hazards and enhancing community adaptation capacities for Vietnam". We also acknowledge additional financial support from the VLIR-UOS TEAM Project (VN2022TEA533A105), "GEOdata infrastructures and citizen SCIences to support REsilient development of rural communities in Quang Nam province (GEOSCIRE)", for completing the publication of this manuscript.

### Competing interests


The contact author has declared that none of the authors has any competing interests.

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
