# Peer review of "Integrating multi-hazard susceptibility and building exposure: A case study for Quang Nam province, Vietnam"

_EGUsphere, 2024_

## Author Comment (AC1)

**List of changes in the revised paper:**

This document explains the changes made in the revised manuscript in order to address comments raised by the reviewers. Reviewers' comments are marked in **black**; authors' response is shown in **blue;** while the changes in the revised manuscript are marked in **red**.

**Response to Reviewers Reviewer #1**

In the study "Integrating multi-hazard susceptibility and building exposure: A case study for Quang Nam province, Vietnam", the authors use an established set of machine learning models to estimate the susceptibility of the Quang Nam province, Vietnam, to floods and wildfires. By creating a comprehensive geospatial database including various historic flood and wildfire events, topographical, geological, hydrological, and climatic features, along with land use and building data, the authors have developed a robust basis for susceptibility mapping regarding the floods and wildfires and offer interesting insights regarding exposure assessment. By combining susceptibility categories for wildfires and floods the study offers a more nuanced perspective on what type of spatially co-occuring multi-hazard events could be expected in different areas of the study region.

The study presents relevant insights into susceptibility factors for floods and wildfires. However, there are several major aspects that require clarification:

**Response:** Thank you for your helpful and detailed comments and suggestions. They have pushed us to improve the manuscript, and we have tried to incorporate it into our revised version. We have carefully read and addressed all comments, point by point, below.

**Specific comment 1:** The authors use the term multi-hazard repeatedly in their study. They introduce the term in line 49 to 51 and indicate that "Multi-hazard susceptibility assessment provides insights into the spatial co-occurrence of multi-hazard" (line 53). Further specification in what type of multi-hazard interactions are relevant for the selected hazard pair of floods and wildfires is not provided. Yet, the study would significantly benefit from such a clarification. For instance, the study could benefit from discussing the dynamic interplay between flood probability in wet seasons and wildfire likelihood in dry seasons. Based on previous studies looking at wildfire-flood interactions, most emphasis has been put on the reduction of infiltration/storage capacity in natural systems that have been burnt (see e.g. Mueller et al. 2018). Similarly, it seems physically plausible that a flood might even reduce the risk of wildfires due to the large-scale wetting of vegetation. I am thus wondering whether the current set-up of this study would rather serve the purpose of multiple hazard susceptibility mapping as it neglects the hazard interaction dynamics that are of critical importance for multi-hazard events? This is particularly critical, since the authors want

to make the step from susceptibility mapping towards exposure mapping. But due to (temporal) dynamics of multi-hazard events, an assumption of constant exposure might be worth discussing.

Mueller, M., Lima, R. E., Springer, A. E., & Schiefer, E. (2018). Using matching methods to estimate impacts of wildfire and post wildfire flooding on house prices. Water Resources Research, 54, 6189–6201. https://doi.org/10.1029/2017WR022195

**Response:** Thanks for your comment. We agree that a multi-hazard analysis is not considered in the sense of interactions between hazards. However, the analysis of the two hazards (floods and fires) in our study is the first step towards a fully multi-hazard risk assessment by looking at where there is a spatial overlap. We explained the dynamic interplay between flood probability in wet seasons and wildfire likelihood in dry seasons in Section 1. Introduction, as follows:

"The term "multi-hazard" refers to the fact that hazards often interact in complex ways, and their combined impact might be greater than the sum of individual hazards (Wing et al. 2018). The dynamic interplay between flood probability in wet seasons and wildfire likelihood in dry seasons can be influenced by various factors, including environmental conditions, climatic patterns, topography, vegetation cover, and land use patterns (Skilodimou et al. 2021; Bountzouklis et al. 2022). Wildfires can significantly impact landscape hydrology by destroying vegetation cover and disrupting soil structure, reducing infiltration rates and heightening surface runoff during subsequent rain events (Mueller et al. 2018). Floods can reduce the formation and expansion of wildfire risks by wetting vegetation and soil, temporarily mitigating the likelihood of ignition and fire spread (Moody and Ebel 2012; Papaioannou et al. 2023). However, flood events can disrupt natural drainage patterns, saturate soils, and promote vegetation development, fueling forest fires in dry seasons (Eisenbies et al. 2007). In general, the formation of multi-hazard events often results from dynamic spatial and temporal interactions among various factors (De Angeli et al. 2022); significantly, floods and wildfires can exacerbate each other's impacts depending on seasonal fluctuations, environmental conditions or extreme climatic variability (Yu et al. 2023). Broadening the assessment framework for these dynamic interactions can lead to a more comprehensive and accurate risk evaluation (De Angeli et al. 2022). Thus, multi-hazard susceptibility and exposure assessments are required for efficient disaster risk management (Zhou et al. 2015). Multi-hazard susceptibility assessment provides insights into the spatial co-occurrence of different hazard types (Rusk et al. 2022). Multi-hazard exposure assessment enables the evaluation of the potential impact of multi-hazards on people, buildings, and critical facilities in a given location (De Angeli et al. 2022). This information is invaluable for emergency response planning, resource allocation, and the development of robust evacuation strategies (Kappes et al. 2012)."

**Specific comment 2:** While the authors do a great job to explain the significance of each of the single-hazard risks in the study area, the importance of the multi-hazard event remains unclear. Also unclear remains, what type of floods the authors are

referring to. In the introduction multiple floods including coastal floods are mentioned. The remainder of this study seems to focus on riverine floods.

**Response:** You are right; the importance of the multi-hazard events requires more complex hydrological modelling to consider the interactions of these two hazards at the catchment scale. We added this limitation to the discussion section.

The flood hazards in the study area are mainly riverine floods. The research area is in Vu Gia-Thu Bon river basin, one of the largest river basins in Vietnam. We revised the manuscript and also explained this problem in Section 1. Introduction, according to your comment as follows:

"Flooding is a common natural hazard in many coastal and low-lying areas worldwide that can be caused by various factors, including heavy rainfall, snowmelt, storm surges, or the breaching of dams and levees (Viglione and Rogger 2015). Quang Nam province is characterized by a coastal and low-lying topography and faces high flood risks due to heavy rainfall, typhoons, and potential breaches of dams and levees (Luu et al. 2018). The province has two large river systems, including the Vu Gia - Thu Bon and Tam Ky rivers. Due to the steep, hilly terrain and prolonged heavy rainfall, the river network of this province is quite dense, and as a result, this province faces riverine floods every year in the lowland area and along the coast. This issue holds particular significance for the Quang Nam province because flood events pose a direct threat to human lives and cause significant damage to its infrastructure, education, economic development, and health-related services (Lee et al. 2020)".

**Specific comment 3:** The authors provide a comprehensive list of study aims. However, I am not sure how targets 1 and 4 are covered in this study. The authors describe well the process of deriving the susceptibility maps, yet there's a lack of evidence considering spatially or temporally co-occurring events where hazard dynamics are relevant. Coming back to the previous comment, multi-hazard events are significant because of their interactions which lead to non-linearly altered impacts. It is also unclear how the outputs of the workflow are assessed to determine whether it is a useful assessment tool and provides decision-support (for what exactly?) regarding risk reduction and management.

**Response:** Thank you for your comments; we revised the target 1 and removed the target 4. The reviewer is right in saying that we model each hazard separately and then look at spatial co-occurrence (combining two different susceptibility maps). We added the limitations into Section 5. Discussion:

"In this study, we have only considered spatially co-occurring multi-hazard events and neglected the dynamic interaction of these hazard events. The obtained exposure maps also need further analysis into the impacts of multi-hazard events to provide more useful information for risk assessment and effectively implement disaster risk management within the study area. A more significant limitation of this research lies in the absence of consideration for stakeholder engagement and feedback while developing and applying

the multi-hazard exposure estimation model. Interaction with stakeholders in charge of risk management would help to identify further the challenges posed by exposure to multi-hazard, validate the modelling approach proposed in this research and specify how the result of such model can best contribute to strengthening the effectiveness of risk management strategies.".

**Specific comment 4:** The method section could benefit from some streamlining. While the overall workflow as presented in Figure 2 is clear and straightforward. Aligning the subsection sequence (and subsection titles) with the workflow presented in Figure 2 would reduce redundancy and improve clarity. Specifically, sections such as 3.3.3 could be embedded in the overall flow (first check multicollinearity, then apply ML model?). Another suggestion would be to present GEE already as part of the methodology flowchart as the overarching framework in which data are combined, the models are set up, tested, and used etc.

**Response:** Thank you for your suggestions. We adjusted the subsection sequence more clearly according to your suggestions.

**Specific comment 5:** The process of combining data for the flood and wildfire inventories could benefit from further elaboration (either in the main text or in a supplement). Regarding floods: a) it seems that the flood events point data and map data were combined. How was this done? How were the points prepared for the combination with the maps for the training? To determine the non-flood points, how were the flood markers considered? b) A specification what made the three flood events historic would be interesting. Are those all the same flood type (e.g. fluvial floods)? For wildfires: a) Which period was considered to determine the 1,911 wildfire locations? Was it just within the last year or the past decade or…? B) When selecting non-wildfire locations, it was assumed that built environments cannot burn. However, when it comes to exposure, to wildfires we would assume that the built environment must be exposed to these fires somehow. It would be helpful if the authors could elaborate in this section (or in the section when describing the built environment), how the choice that built environments are non-wildfire locations influence the outcomes of the machine learning training to spot fires that endanger built environment. Also, it would be interesting to learn whether there are any multi-hazard events in the dataset of historic events (potentially to be added to the supplement?)?

**Response:** We thank the reviewer for the excellent comments. We agree with the reviewer that this is a bit weird and can lead to reducing the estimated exposure of build-up area to burn. We explained as follows:

* Regarding floods:

We combined the flood locations from 2 sources:

1) 847 historical flood marks of 2007, 2009, and 2013 floods. This data was investigated by the Quang Nam Provincial Steering Committee of Natural Disaster Prevention and

Control in 2013 and JICA in 2007 and 2009. These points were recorded with specific measurements of flood depth.

2) 47 new flood locations were detected from Sentinel 1 during 2017-2021. We get 47 points at the centroid of 47 flood area polygons. In reviewing your comments, we have checked all 2007, 2009 and 2013 flood points, and they are also within the 2017-2021 flood zone. Additionally, the non-flood points were determined by overlaying all flood inventory onto the study area. We explained these problems quite clearly in Subsection 3.2.1. Inventories of floods and wildfires are as follows:

"The development of a reusable and accurate flood inventory is a crucial element in the susceptibility mapping of floods (Ahmadlou et al., 2018). In this study, the flood marker points recorded for all floods from 2007 to 2023 were considered. This data was obtained from the Quang Nam Provincial Steering Committee of Natural Disaster Prevention and Control. Data cleaning to remove duplicates resulted in 847 historical flood marks, corresponding mainly to the major floods of 2007, 2009, and 2013. Each flood mark comprises a unique identifier, geographical coordinates (longitude and latitude), flood depth, and provider information. A second source of information was derived from mapping flood extent on SAR data from Sentinel 1 for 2017 to 2023, which we compare with official reports from the Provincial Committee. We coded in Google Earth Engine to detect flood areas as in this study (Mai Sy et al., 2023). All inundation areas for several years were overlayered and compared with the flood mark locations to avoid duplicates. Forty-seven new flood areas were detected and integrated with 847 historical flood marks for the inventory data.

The final flood inventory includes 894 flood locations: 70% of them (626 locations) were randomly selected to calibrate the flood susceptibility model, and the remaining 30% (268 locations) were designated for validating purposes (Figure 3). In addition, a total of 894 non-flood locations were randomly selected across the study area using the "Create random point tool" in ArcGIS software. The non-flood points were determined by overlaying all flood inventory onto the study area map. Non-flood points were only selected in zones that were outside all flood maps. Additionally, we excluded steep slopes (>10°) or areas of positive relief (such as hilltops) from the selection of non-flood points, as these locations that can evidently not be associated with floods would artificially increase the accuracy of the susceptibility model. The non-flood points were then classified in a ratio of 70/30, mirroring the classification of the flood locations. This process was undertaken to create a comprehensive database for input into the GEE platform, which was utilized for modelling and validation."

* For wildfires:

In this study, 1,911 wildfire locations were collected from March to August 2020 to 2023 and provided by the National Forest Protection Department (available at https://watch.pcccr.vn/thongKe/diemChay). The periods of data collection correspond to the study area's dry season, from March to August every year. We added this information in Subsection 3.2.1. Inventories of floods and wildfires are as follows:

"For the wildfire inventory, this study involved the collection of 1,911 wildfire locations from March to August 2020 to 2023 (**Figure 3**), corresponding to the peak of the dry season, from the National Forest Protection Department's website (available at https://watch.pcccr.vn/thongKe/diemChay). This agency utilizes data from many satellites (AQUA, J1, SUOMI, and TERRA) that are regularly received at the TerraScan receiving station located at the Forest Protection Department. The use of near-infrared bands from many satellites helps to identify the presence of heat associated with active fires on the ground (Giglio et al. 2008). Changes in temperature and environment (humidity, wind direction) on the ground on the same day have been determined and duplicated at one fire location based on near-infrared bands of types of satellites to identify fire hotspots. Normally, these changes are observed on the same day and then matched with data from near-infrared bands on specific types of satellites to identify active fire hotspots (Kharyutkina et al. 2022; Marinho et al. 2021). The National Forest Protection Department website database was checked and filtered to avoid duplicated wildfire locations, dates, and commune data field conditions. The wildfire location data represents the specific fire sites captured by one type of satellite inside a particular commune at a given time. The area criterion is also important for obtaining accurate natural fire locations and eliminating human-made ones. We used a filtering method to find wildfire spots that were larger than 2 hectares, using information noted in the statistical data provided by the National Forest Protection Department. To determine the non-fire point, we randomly selected points within the zones with forested and natural vegetation land cover, which were not identified as wildfires in the inventory. We excluded residential areas, water systems, and crop areas from the selection of non-fire points, as these cannot be associated with wildfires corresponding to the criteria selected in this study and would artificially increase the accuracy of our susceptibility model.

We agree with you that although built environments are not typically considered burn points, they can still be exposed to wildfires in various ways. However, Quang Nam is a central coastal province of Vietnam; the population is concentrated in the coastal plain, along National Highway 1A, Vu Gia Thu Bon and Tam Ky plains. In addition, the selected non-fire points have been verified and confirmed through field surveys and local authorities.

**Specific comment 6:** The authors comprehensively describe what influencing factors are considered. For some of the influencing factors with temporal and/or spatial variability (e.g. precipitation, temperature) it is unclear how the collected data are further processed. For example, precipitation are collected for a period of 10 years, while the

considered flood marker cover the periods 2007, 2009, 2013 and 2017-2021. The same applies for the temperature data where data were available only for 3 years. How were the influencing factors considered for wildfires that took place outside of this period (assuming that this is the case, since no specification is made with regards to the time horizon over which the wildfire locations were collected). Same holds for the NDVI index, where it is not clear when this imagery has been produced. Additionally, it would be valuable if the authors could reflect on the interpolation method used to inter/extrapolate between the gauge stations. Are the gauges distributed sufficiently well and do the elevation/similarity characteristics allow for the application of the chosen method?

**Response:** For some of the influencing factors with temporal and/or spatial variability (e.g. precipitation, temperature), we used the IDW interpolation method to extract the rainfall and temperature for the study area. In the manuscript, we explained the collected time and processing technique for these data:

"Daily rainfall data was recorded from 2003 to 2023 and collected from 33 rain gauge stations in Quang Nam province. This study used the *Inverse Distance Weighted* technique to separately generate average yearly cumulated rainfall maps for the rainy and dry seasons."

and

"The daily temperature data were collected from March to August between 2020 and 2023 (dry seasons) at https://power.larc.nasa.gov/data-access-viewer/. This research used the Inverse Distance Weighted approach to produce a temperature map specifically for dry seasons (March to August).".

In this study, the historical flood marker points have been considered continuously from 2007 to 2023. However, we collected, synthesized, and removed duplicate flood points, so there were 847 historical flood marks of 2007, 2009, and 2013 historical flood events obtained from the Quang Nam Provincial Steering Committee of Natural Disaster Prevention and Control and 47 flood points explored from Sentinel 1 for 2017 to 2023. We added the information into Subsection 3.2.1 Inventories of floods and wildfires:

"In this study, the flood marker points were considered for all flood events from 2007 to 2023 as reported by the Quang Nam Provincial Steering Committee of Natural Disaster Prevention and Control. We removed duplicate flood points. A total of 847 historical flood marks were obtained from this database – these correspond mainly to the 2007, 2009, and 2013 flood events with the largest spatial extent. Each flood mark comprises a unique identifier, geographical coordinates (longitude and latitude), flood depth, and provider information. A second source of information was derived from mapping flood extent on SAR data from Sentinel 1 for 2017 to 2023, which we compare with official reports from the Provincial Committee. We coded in Google Earth Engine to detect flood areas as in this study (Mai Sy et al. 2023). After that, the inundation areas of many years were overlayed and compared with the flood mark locations to avoid duplicates.

47 new flood sites were detected and integrated with 847 historical flood marks for the inventory data.".

We also explained the NDVI calculation for this research: "This study calculated the NDVI index from the Landsat 8 imagery. The NDVI index is the average value for the rainy and dry seasons separately from 2020 to 2023 – the same period over which the fire dataset is available.".

We revised the NDVI index for flood hazard and wildfire hazard modelling. The NDVI for flood hazard is from September to February (rainy season), and wildfire hazard is from March to August (dry season) as follows:

[Figure]

**Specific comment 7:** While the authors do well in qualitatively describing the algorithm and underlying principles, key information regarding the hyperparameter tuning (and final chosen ones) and pruning techniques are not provided. As such it is difficult to reproduce the results. The results of the tests for the hyperparameter tuning could be useful additions as supplemental information. On a similar note, further quantitative information regarding the bootstrapping (e.g. number of bootstrapped samples) could be relevant as well. Furthermore, it would be helpful for readers less familiar with ML methodology (like me) to link the parameters used in the equations to the inputs used in this specific study. For example, what are D, N, X and Y in the CART?

**Response:** We thank the reviewer's comment. We explained the used parameters in the equations in Subsection 3.3.2 Machine learning approach for hazard susceptibility modelling. Additionally, we also explained key information regarding the bootstrapping, the hyperparameter tuning (and final chosen ones), and pruning techniques in Subsection 3.4 Experimental process, as follows:

3.4 Experimental process

This study employed the GEE cloud computing platform for the pixel-based CART and RF algorithms to build susceptibility maps for flood and wildfire hazards separately. The input data was collected from various sources and formats. First, we pre-processed and converted these data into raster format with 30-meter spatial resolution in a GIS environment. Then, these data were uploaded into the GEE platform. The following processing steps were followed:

1. Data preparation: All thematic maps were uploaded, overlaid, and gathered into each dataset for modelling flood and wildfire susceptibility. Inventory maps of floods and wildfires were uploaded and linked with the corresponding attributes. These inventory maps were classified into two datasets: the training dataset, with 70% inventory locations, and the testing dataset, with 30% remaining inventory locations.

2. Hyperparameter tuning: The CART and RF models were developed to construct each hazard susceptibility map on the training dataset. The hyperparameter tuning process was carried out for both ML algorithms, in which the parameter space for each ML algorithm was identified (**Table 1**). The CART algorithm included factors like *max depth*, *min samples split*, *min samples leaf*, and *criterion*. The RF algorithm encompassed the *number of trees* (*n_estimators*), *max features*, *max depth*, *min samples split*, and *min samples leaf*. Hyperparameter tuning techniques such as grid or random search were applied to explore and evaluate different parameter combinations systematically.

3. Model training: The obtained optimal hyperparameters from the tuning process were utilized to train the predictive models.

4. Pruning techniques: Pruning techniques were employed to reduce overfitting by refining tree structures based on complexity measures or validation performance for the CART algorithm. For the RF algorithm, these techniques were applied to control hyperparameters like *maximum tree depth* and *minimum samples per leaf*, helping adjust model complexity.

5. Model evaluation: The ROC curve and AUC value were applied to validate and compare the predictive performance of each hazard susceptibility model on the validation dataset to select the best predictive model for each hazard.

**Table 1** The parameters of running models in GEE.

| Parameter | Model | | |
| --- | --- | --- | --- |
| | CART | RF | Legend |
| Number of Trees | Null | 200 | The number of decision trees to create. |
| Variables per split | Null | Null | The number of variables per split. If unspecified, uses the square root of the number of variables. |

| | | | |
|---|---|---|---|
| Min leaf population | 1 | 1 | Only create nodes whose training set contains at least this many points. |
| Bag fraction | Null | 0.1 | The fraction of input to bag per tree. |
| Max nodes | 150 | null | The maximum number of leaf nodes in each tree |
| Seed | Null | 23 | The randomization seed |

**Specific comment 8:** The discussion section could benefit from some critical reflection on the decisions made in this study set-up and discuss some of the limitations that come with it. This is particularly critical since the authors claim, that this workflow could be extended by including different hazards and applied in different regions. For example, with reference to Line 458: Both in terms of inputs as well as in terms of how multi-hazard has been defined and conceptualized. The aspect of dynamics has been neglected and it has mostly been looked at spatially co-occurring (without temporal memory) hazard events. Or with reference to Line 489: From the results it seemed that multi-hazard seems to be a less prominent problem (both in terms of susceptibility as well as the exposure). So, a planner could also read the results as mentioned by the authors: "flood risk is much more of a problem, we should focus on that!". I would suggest specifying that with these exposure maps, further analysis into the impacts of multi-hazard events can be made that ultimately can inform multi-hazard risk assessment and thus effective DRM.

**Response:** We thank the reviewer's comment, we agree that this research still lacks evidence of interactions between forest fires and floods in the study region. We cannot confirm that there have been more floods in the years with many wildfires or in the catchments that have experienced large fires. One issue overlooked so far is that wildfires and floods do not have to occur at the same location to interact - burned areas can happen in the upland and generate/influence floods in the lower basins. We added the limitations of this research in Section 5. As per your advice, the discussion follows:

"Considering the spatial occurrence of hazards and the associated exposure to build-up environment enables highlighting which areas and which proportion of buildings are exposed to one specific hazard or both, which can already be relevant for risk management. To consider temporal relationships between hazards (i.e. fire during the dry season inducing flood in the next rain season) or non-local dynamic interactions (i.e. wildfire in upper catchment increasing flood occurrence downstream) would require more process-oriented hazard modelling at a more local scale. A more significant limitation lies in the absence of consideration for stakeholder engagement and feedback while developing and applying the multi-hazard exposure estimation model. This engagement process would validate our model, foster a more comprehensive

understanding of multi-hazard exposure, and strengthen the effectiveness of risk management strategies in real-world scenarios.".

**Specific comment 9:** Have the authors considered to deposit input data maps, algorithms, and model code in FAIR-aligned repositories/archives in alignment with the ambition of NHESS to support open data?

**Response:** Thanks for your comment. We will provide input data maps, algorithms, and model code supporting the findings of this study according to your comment.

**Specific comment 10:** Minor comments

1. The introduction generally includes all relevant elements. The overall story for the introduction could be refined, e.g. by avoiding duplication (compare lines 91 to 99 with lines 32 to 45). Similarly, lines 60 to 88 provide in depth introduction to the ML and previous practice. At the same time, the authors mention multiple information which are quite interesting, but seem to be not relevant for this study (e.g. line 61 to 63; 63 to 65; 68 to 70). I would also suggest trying to integrate lines 60 to 77 with the current practice described in lines 78 to 88).

**Response:** We improved paragraphs in the Introduction section to avoid duplication.

2. The methodological flow is described nicely. However, it seems that there is a lot of overlap in lines 135 to 139 compared to line 139 to 147. Streamlining the text could help the reader.

**Response:** We rewrote the text in the Subsection 3.1 Methodology flowchart to avoid duplication.

3. Figure 2: The flowchart is very nice. Couple of questions:
   - What is the importance of different colors used in this figure? I tried to understand why certain boxes were colored in different colors (e.g. flood influencing factors vs floods, same colors for e.g. ML vs Testing…). If there is a reason for specific colors, I would suggest making it clearer (e.g. explaining in the figure description) or otherwise reduce the number of colors used.
   - I was expecting that the susceptibility maps would be built after the validation exercise. The flow suggest that they were created directly from the training dataset?

**Response:** We reduced the number of colors used in the flowchart and adjusted the order to build susceptibility maps.

4. Line 174 to 175: I don't understand this sentence. Is that the method to determine whether a wildfire has occurred?

**Response:** That is the method to determine whether a wildfire has occurred in the Quang Nam province by the National Forest Protection Department (available at https://watch.pcccr.vn/thongKe/diemChay)

5. Line 176 to 177: This sentence seems unclear to me. What filter has been applied to filter what?

**Response:** A file with the Excel format was explored and downloaded from the website of the National Forest Protection Department (available at https://watch.pcccr.vn/thongKe/diemChay), which is collected from many satellite image sources, so it was necessary to check and filter to avoid duplicated wildfire locations, dates, and positions.

6. Line 180: it is not clear whether areas larger than 2 ha were assumed to be human caused.

**Response:** Wildfire areas smaller than 2 ha were interpreted as induced by human activity based on annotations provided in the statistical data of the National Forest Protection Department. This information was represented in our manuscript as follows:

"We used a filtration process only to retain wildfire spots that exceed a minimum size threshold of 2 hectares, as smaller fire areas should be considered human-induced according to the National Forest Protection Department."

7. Figure 4: I would suggest to either add a bit more text to explain the different maps as part of the influencing factors or place Figure 4 in the appendix. In the appendix, individual plots could also be resized so that legends are better readable.

**Response:** We moved Figure 4 to the appendix.

8. Line 188: How was this set of influencing factors determined? For flooding, proximity to coast could also be a determinant of (coastal) flooding?

**Response:** The set of influencing factors was determined based on their relevance and the data availability within the research area. In this study, we mentioned riverine floods in the Quang Nam province, so we did not use the proximity to the coast.

9. Line 301 to 302: What does these choices of filtering for confidence interval mean? What type of buildings are more likely to be disregarded with the chosen confidence intervals?

**Response:** The type of building and confidence intervals are presented in the study of Sirko et al. (2021) (https://arxiv.org/abs/2107.12283). We cited this work in the manuscript. We also based on the province population to find the appropriate confidence intervals for the research area. The population is 1.5 million, and the total buildings with 80% confidence intervals are 442,220. Assume that about 3-4 people per

building on average is realistic. We also randomly manually check building accuracy on Google Earth for the appropriate confidence intervals.

10. Line 354 to 366: This section seems almost identical with the workflow presented and discussed alongside figure 2. I would suggest streamlining the method section and remove Section 3.2.2 and add relevant information in previous sections. For example, the information that CART and RF work cell-based is quite a relevant information given that the flood and wildfire inventories are point information.

**Response:** We agree with you. We removed Subsection 3.2.2 and added relevant information to explain how CART and RF models work.

11. Line 386: Can the authors explain how the importance sampling can inform which factors have the highest impact on multi-hazard formations? The algorithm used is applied to single hazards (either floods or droughts) but not the multi-hazards?

**Response:** This is the relative importance of variables in modelling single-hazard and not multiple-hazards.

12. Figure 5: Aligning terminology (either testing or validating dataset) would help the readability. Also, what does the Se stand for?

**Response:** We fixed all texts with training and testing datasets. We added the note in the figures that "Se" term stands for standard error.

13. Line 453 to 455: Can the authors clarify what they mean when they claim that floods and wildfires have 'similar spatial extent' and frequency?

**Response:** In this study, we synthesized and collected flood and wildfire inventory based on historical data, satellite imagery, and reports from relevant authorities. While the frequency of floods and wildfires occurs separately, the occurrence of both hazards depends on the season or year.

14. Line 483 to 485: How is this finding affected by the choice to define built-up areas as non-wildfire areas when creating the training data set? It seems that seeing less fires in built up areas could also be influenced by the fact that the ML algorithms were taught that wildfires just don't occur in more densely populated areas?

**Response:** We agree with you. We do NOT select non-fire points in the build up environment.

15. Line 486: How do the authors derive the claim, that the chosen method works well with recurring hazard events? The applied methods seemed not to account for the changes in the physical system induced by either floods or wildfires.

**Response:** We thank the reviewer for your excellent comment. We added this limitation into Section 5. Discussion: "Our findings suggest that ML models such as CART and RF should be used to analyze multi-hazard exposure for various geographical areas particularly susceptible to recurring incidents of wildfire and floods. Our data has shown these tools to model risk and exposure effectively. However, the applied methods in this study did not account for the changes in the physical system induced by either floods or wildfires.".

**References**

Bountzouklis, C., Fox, D. M., and Di Bernardino, E.: Environmental factors affecting wildfire-burned areas in southeastern France, 1970–2019, Natural Hazards and Earth System Sciences, 22, 1181-1200, 10.5194/nhess-22-1181-2022, 2022.

De Angeli, S., Malamud, B. D., Rossi, L., Taylor, F. E., Trasforini, E., and Rudari, R.: A multi-hazard framework for spatial-temporal impact analysis, International Journal of Disaster Risk Reduction, 73, 102829, 10.1016/j.ijdrr.2022.102829, 2022.

Eisenbies, M. H., Aust, W. M., Burger, J. A., and Adams, M. B.: Forest operations, extreme flooding events, and considerations for hydrologic modeling in the Appalachians—A review, Forest Ecology and Management, 242, 77-98, 10.1016/j.foreco.2007.01.051, 2007.

Giglio, L., Csiszar, I., Restás, Á., Morisette, J. T., Schroeder, W., Morton, D., and Justice, C. O.: Active fire detection and characterization with the advanced spaceborne thermal emission and reflection radiometer (ASTER), Remote Sensing of Environment, 112, 3055-3063, 10.1016/j.rse.2008.03.003, 2008.

Kappes, M. S., Keiler, M., Von Elverfeldt, K., and Glade, T.: Challenges of analyzing multi-hazard risk: a review, Natural hazards, 64, 1925-1958, 10.1007/s11069-012-0294-2, 2012.

Kharyutkina, E., Pustovalov, K., Moraru, E., and Nechepurenko, O.: Analysis of spatio-temporal variability of lightning activity and wildfires in western Siberia during 2016–2021, Atmosphere, 13, 669, 10.3390/atmos13050669, 2022.

Lee, J., Perera, D., Glickman, T., and Taing, L.: Water-related disasters and their health impacts: A global review, Progress in Disaster Science, 8, 100123, https://doi.org/10.1016/j.pdisas.2020.100123, 2020.

Luu, C., Von Meding, J., and Kanjanabootra, S.: Flood risk management activities in Vietnam: A study of local practice in Quang Nam province, International Journal of Disaster Risk Reduction, 28, 776-787, https://doi.org/10.1016/j.ijdrr.2018.02.006, 2018.

Mai Sy, H., Luu, C., Bui, Q. D., Ha, H., and Nguyen, D. Q.: Urban flood risk assessment using Sentinel-1 on the google earth engine: A case study in Thai Nguyen city, Vietnam, Remote Sensing Applications: Society and Environment, 31, 10.1016/j.rsase.2023.100987, 2023.

Marinho, A. a. R., Gois, G. D., Oliveira-Júnior, J. F. D., Correia Filho, W. L. F., Santiago, D. D. B., Silva Junior, C. a. D., Teodoro, P. E., De Souza, A., Capristo-Silva, G. F., Freitas, W. K. D., and Rogério, J. P.: Temporal record and spatial distribution of fire foci in State of Minas Gerais, Brazil, Journal of Environmental Management, 280, 111707, 10.1016/j.jenvman.2020.111707, 2021.

Moody, J. A. and Ebel, B. A.: Hyper-dry conditions provide new insights into the cause of extreme floods after wildfire, CATENA, 93, 58-63, 10.1016/j.catena.2012.01.006, 2012.

Mueller, J. M., Lima, R. E., Springer, A. E., and Schiefer, E.: Using matching methods to estimate impacts of wildfire and postwildfire flooding on house prices, Water Resources Research, 54, 6189-6201, 10.1029/2017WR022195, 2018.

Papaioannou, G., Alamanos, A., and Maris, F.: Evaluating Post-Fire Erosion and Flood Protection Techniques: A Narrative Review of Applications, GeoHazards, 4, 380-405, 10.3390/geohazards4040022, 2023.

Rusk, J., Maharjan, A., Tiwari, P., Chen, T.-H. K., Shneiderman, S., Turin, M., and Seto, K. C.: Multi-hazard susceptibility and exposure assessment of the Hindu Kush Himalaya, Science of The Total Environment, 804, 150039, https://doi.org/10.1016/j.scitotenv.2021.150039, 2022.

Skilodimou, H. D., Bathrellos, G. D., and Alexakis, D. E.: Flood hazard assessment mapping in burned and urban areas, Sustainability, 13, 4455, 10.3390/su13084455, 2021.

Viglione, A. and Rogger, M.: Chapter 1 - Flood Processes and Hazards, in: Hydro-Meteorological Hazards, Risks and Disasters, edited by: Shroder, J. F., Paron, P., and Baldassarre, G. D., Elsevier, Boston, 3-33, https://doi.org/10.1016/B978-0-12-394846-5.00001-1, 2015.

Wing, O. E. J., Bates, P. D., Smith, A. M., Sampson, C. C., Johnson, K. A., Fargione, J., and Morefield, P.: Estimates of present and future flood risk in the conterminous United States, Environmental Research Letters, 13, 10.1088/1748-9326/aaac65, 2018.

Yu, G., Liu, T., Mcguire, L. A., Wright, D. B., Hatchett, B. J., Miller, J. J., Berli, M., Giovando, J., Bartles, M., and Floyd, I. E.: Process-Based Quantification of the Role of Wildfire in Shaping Flood Frequency, Water Resources Research, 59, e2023WR035013, 10.1029/2023WR035013, 2023.

Zhou, Y., Liu, Y., Wu, W., and Li, N.: Integrated risk assessment of multi-hazards in China, Natural hazards, 78, 257-280, 10.1007/s11069-015-1713-y, 2015.

---

## Author Comment (AC2)

**List of changes in the revised paper:**

This document explains the changes made in the revised manuscript while adressing the comments raised by the reviewer. Reviewers' comments are marked in **black**; authors' response is shown in **blue;** while the changes in the revised manuscript are marked in **red**.

**Response to Reviewers Reviewer #2**

**General comment:** While the paper is generally well-written, there could be improvements in organizing the content to enhance readability and flow, particularly in presenting the methodology and results sections.

**Response:** We thank the reviewer for helpful and detailed comments and suggestions. They contributed to further improving our manuscript. The methodology section has been significantly restructured to avoid duplication and improve readability.

**Specific comment 1:** It would be beneficial to include a more detailed discussion on the validation process and uncertainty analysis of the models to ensure the robustness and reliability of the findings.

**Response:** The validation is embedded in our susceptibility model (but does not account for the exposure aspect). There are several sources of uncertainty, including models and exposure data. Future research might modify and enhance the exposure and susceptibility groups.

**Specific comment 2:** It is not entirely clear from the paper how the multiple hazards (floods and wildfires) are integrated into the multi-hazard exposure estimation. The methodology section should provide a more detailed explanation of the approach used to combine and assess the compound risk arising from different hazards. Clarifying this aspect would help readers better understand the synergistic effects of multiple hazards and how they contribute to overall risk.

**Response:** Thanks for your comment. We clarify that the hazard susceptibility maps were produced separately for flood and wildfires, and then combined in a multi-hazard susceptibility maps that consider only the spatial co-occurrence of these hazards, without considering dynamic interactions. We explained the approach used to combine and assess the compound risk arising from different hazards in Section 3.1 Methodology flowchart, as follows:

**3.1 Methodology flowchart**
The implementation process comprises seven main stages, as follows: (1) Factors potentially influencing the spatial distribution of floods and wildfire were collected, including topography, geology, hydrology, climate (temperature, wetness, wind), and land use based on their relevance and data availability (Luu et al. 2018; Pham et al. 2021), (2) Inventory maps of each hazard were created based on historical data

collection, (3) The influencing factors of each hazard were tested for multicollinearity to enhance the reliability and stability of the model's predictions, (4) CART and RF models were developed on the GEE cloud computing platform to construct susceptibility maps of floods and wildfires separately, (5) The Area Under the ROC Curve (hereafter, AUC) was utilised to assess the predictive performance of the susceptibility map to choose the best model for each hazard and validate it, (6) The flood susceptibility map and the wildfire susceptibility map were combined to build a multi-hazard susceptibility map, and (7) this multi-hazard susceptibility map was overlaid with the building data to create a multi-hazard exposure map for the study area (**Figure 2**).

[Figure]

Figure 1. Methodology flowchart for multi-hazard exposure assessing and mapping in this study.

**Specific comment 3:** My previous comment is of special relevance when the two hazards analyzed are common to happen in different hydrological seasons. Exploring the interactions, dependencies, and cumulative effects of floods and wildfires would provide valuable insights into the complex nature of multi-hazard scenarios. A comparative analysis of the combined risk versus individual hazards would further highlight the significance of considering multiple hazards in risk assessment and management.

**Response:** We agree with you. To highlight the significance of considering multiple hazards in risk assessment and management, we explained the dynamic interplay between flood probability in wet seasons and wildfire likelihood in dry seasons in Section 1. Introduction. We acknowledge that your assessment is limited to spatial co-occurrence without considering temporal links or dynamic interactions, so we added the limitations into Section 5. Discussion: "In this study, we have only considered spatially

co-occurring multi-hazard events and neglected the dynamic interaction of these hazard events. The obtained exposure maps also need further analysis into the impacts of multi-hazard events to provide more useful information for risk assessment and effectively implement disaster risk management within the study area. A more significant limitation of this research lies in the absence of consideration for stakeholder engagement and feedback while developing and applying the multi-hazard exposure estimation model. Interaction with stakeholders in charge of risk management would help to identify further the challenges posed by exposure to multi-hazard, validate the modelling approach proposed in this research and specify how the result of such model can best contribute to strengthening the effectiveness of risk management strategies.".

**Specific comment 4:** Consideration of stakeholder engagement and feedback in the development and application of the multi-hazard exposure estimation model could enhance the relevance and applicability of the research to real-world scenarios.

**Response:** We agree that consultation with the stakeholders is very important. We will consider stakeholder engagement and feedback in the next stage of our GeoSciRe project. At this stage, we only present some potential results and approaches in this paper. We added this limitation to the Discussion section as follows:

"In this study, we have only considered spatially co-occurring multi-hazard events and neglected the dynamic interaction of these hazard events. The obtained exposure maps also need further analysis into the impacts of multi-hazard events to provide more useful information for risk assessment and effectively implement disaster risk management within the study area. A more significant limitation of this research lies in the absence of consideration for stakeholder engagement and feedback while developing and applying the multi-hazard exposure estimation model. Interaction with stakeholders in charge of risk management would help to identify further the challenges posed by exposure to multi-hazard, validate the modelling approach proposed in this research and specify how the result of such model can best contribute to strengthening the effectiveness of risk management strategies."

**References**

Luu, C., Von Meding, J., and Kanjanabootra, S.: Assessing flood hazard using flood marks and analytic hierarchy process approach: a case study for the 2013 flood event in Quang Nam, Vietnam, Natural Hazards, 90, 1031-1050, https://doi.org/10.1007/s11069-017-3083-0, 2018.
Pham, B. T., Luu, C., Phong, T. V., Nguyen, H. D., Le, H. V., Tran, T. Q., Ta, H. T., and Prakash, I.: Flood risk assessment using hybrid artificial intelligence models integrated with multi-criteria decision analysis in Quang Nam Province, Vietnam, Journal of Hydrology, 592, https://doi.org/10.1016/j.jhydrol.2020.125815, 2021.

---

## Referee Report (RR1)

The paper is well written, and the changes implemented are in the direction of the comments of the previous reviewers. In general, the paper applies two Machine Learning methods (CART and RF) to produce a multi-hazard susceptibility map in the Region of Quang Nam (Vietnam) for floods and wildfires, overlaying the results to create an exposure map for buildings. It describes a robust methodology for spatial co-occurring multi-hazard susceptibility maps, from the creation of a geospatial database of historical wildfires and floods events to the choice of susceptibility factors and the training/testing of ML algorithms for single hazard susceptibility maps. The addition of a building exposure layers to the multi-hazard susceptibility maps is one step towards a multi-risk analysis that can be used to inform used to inform local communities and regulatory authorities. However, there are some aspects that still require some clarification:

Specific Comment 1:
The authors added insights on the limitations of the multi-hazard susceptibility mapping, clarifying that the focus of the paper is on spatially co-occurring hazards, and that it does not delve into the analysis of the dynamical interactions between the various hazards. However, it might be important to also discuss more about the choice of exposure layers, (as also stated by Reviewer 1, Comment 1 *"an assumption of constant exposure might be worth discussing"*).  In particular, one important aspect in extending a multi-hazard to an analysis of risk for a specific asset (such as buildings) is the role of vulnerability, which is never mentioned in the paper. This is critical because, for example, the characteristics of buildings might make them more resilient towards one hazard, but more at risk for the second one. The choice of a constant exposure layer for both hazards should then be discussed, and potential limitations/future developments clearly stated.

Specific Comment 2:
There are multiple references to landslide susceptibility mapping, but the paper focuses (and trains ML models) only on wildfires and flood susceptibility mapping. In particular, Chapter 3.2.2 should be changed, to discuss the factors that are relevant for wildfires and floods (now only in the Supplementary material). Also, in Chapter 3.4 ("Experimental process") there are some references to landslide susceptibility, which should be removed. If landslide susceptibility is to be discussed, it should be mentioned in the discussion chapter, as a possible extension.

Specific Comment 3:
The formula for GINI impurity (Line 218) is wrong: the factor is the sum of pairwise products of the probabilities for each class, thus the correct formula should be:
$$\sum_{i=1}^{J} P_i \left(1 - P_i\right) = \sum_{i=1}^{J} P_i - \sum_{i=1}^{J} P_i^2 = 1 - \sum_{i=1}^{J} P_i^2$$

Moreover, it could be useful for assessing the robustness of the hyperparameter tuning to know which was the range the range of the hyperparameters tested in the Cross Validation and not only the final selection (Table 1), either in the main chapter or in the supplementary materials.

---

## Author Response (AR2)

**Response to Reviewers Reviewer #1**

**Specific comment 1:** I want to thank the authors for their thorough revision and comprehensive response to my questions. Most of them have been solved. The only remaining question is regarding the title of the study. The authors confirm in their reply that they do not account of multi-hazard characteristics but make a first, crucial step towards multi-hazard susceptibility mapping by developing multiple hazard susceptibility maps. I would suggest that the authors reflect on the title and align it with the key findings/approach of the study (and make some minor adjustments in the introduction, discussion and conclusion sections where necessary).

I enjoyed reviewing this interesting study, learned many new things and hope the authors found the feedback useful and constructive.

**Response:** We thank you for your kind words and appreciate your insightful comments and suggestions throughout the review process.

Regarding your suggestion on the title of the study, we agree that aligning the title with the focus of our study is important. While our research makes a significant step towards multi-hazard susceptibility mapping, it does not fully address the dynamic interactions between hazards.

We revised the title accordingly to emphasize that the paper primarily develops susceptibility maps for multiple co-occurring hazards rather than fully integrating multi-hazard characteristics.

The revised title is "Integrating susceptibility maps of multiple hazards with building exposure: A case study of wildfires and floods in Quang Nam province, Vietnam".

**Response to Reviewers Reviewer #2**

**General comment:** The paper is well written, and the changes implemented are in the direction of the comments of the previous reviewers. In general, the paper applies two Machine Learning methods (CART and RF) to produce a multi-hazard susceptibility map in the Region of Quang Nam (Vietnam) for floods and wildfires, overlaying the results to create an exposure map for buildings. It describes a robust methodology for spatial co-occurring multi-hazard susceptibility maps, from the creation of a geospatial database of historical wildfires and floods events to the choice of susceptibility factors and the training/testing of ML algorithms for single hazard susceptibility maps. The addition of a building exposure layers to the multi-hazard susceptibility maps is one step towards a multi-risk analysis that can be used to inform used to inform local communities and regulatory authorities. However, there are some aspects that still require some clarification:

**Response:** Thank you for your positive evaluation of our work and for acknowledging the improvements made based on previous reviewer comments. We are grateful for your constructive feedback, and we will address these aspects thoroughly in the revised version of the paper.

**Specific comment 1:** The authors added insights on the limitations of the multi-hazard susceptibility mapping, clarifying that the focus of the paper is on spatially co-occurring hazards, and that it does not delve into the analysis of the dynamical interactions between the various hazards. However, it might be important to also discuss more about the choice of exposure layers, (as also stated by Reviewer 1, Comment 1 "an assumption of constant exposure might be worth discussing"). In particular, one important aspect in extending a multi-hazard to an analysis of risk for a specific asset (such as buildings) is the role of vulnerability, which is never mentioned in the paper. This is critical because, for example, the characteristics of buildings might make them more resilient towards one hazard, but more at risk for the second one. The choice of a constant exposure layer for both hazards should then be discussed, and potential limitations/future developments clearly stated.

**Response:** We thank the reviewer's insightful comment regarding the need to discuss the choice of exposure layers and the role of vulnerability in the multi-hazard susceptibility mapping. We acknowledge the importance of incorporating vulnerability in extending the analysis from hazard susceptibility to risk, especially in the context of specific assets like buildings, which can exhibit varying resilience depending on the hazard type. In response, we added the explain for chosing the building is a primary exposure layer in section 3.2.3 as follows:

"This study focuses on buildings in terms of elements exposed to a hazards, considering their importance as critical economic assets and reflections of population

distribution (Askar et al. 2021). Buildings are essential components of community infrastructure, and damage to them may have big social and economic effects, making them a crucial exposure indicator for risk assessment (Carreño et al. 2007). In addition, buildings often accommodate individuals and vital services; thus, their exposure to hazards and susceptibility to damage  directly control the possibility of human fatalities and disturbance to everyday activities. In terms of vulnerability, buildings are not equally at risk from all hazards; their susceptibility varies depending on the hazard type and the structural characteristics of the building, although vulnerability is not considered explicitly in this study (Schneiderbauer and Ehrlich 2004)"

**Specific comment 2:** There are multiple references to landslide susceptibility mapping, but the paper focuses (and trains ML models) only on wildfires and flood susceptibility mapping. In particular, Chapter 3.2.2 should be changed, to discuss the factors that are relevant for wildfires and floods (now only in the Supplementary material). Also, in Chapter 3.4 ("Experimental process") there are some references to landslide susceptibility, which should be removed. If landslide susceptibility is to be discussed, it should be mentioned in the discussion chapter, as a possible extension.

**Response:** Thank you for highlighting the inconsistency regarding the references to landslide susceptibility mapping. We fully acknowledge that the paper focuses on wildfire and flood susceptibility mapping and that including landslide susceptibility references may cause confusion. In response, we have made the following revisions:

Chapter 3.2.2: We have revised this section to focus exclusively on the factors relevant to wildfires and floods, ensuring that this section aligns with the central objectives of the paper as follows:

**3.2.2 Influencing factors**

Several factors significantly influence flood and wildfire occurrences. Low-lying areas are prone to flooding, while elevated regions can hinder fires (Pourtaghi et al. 2016; Bui et al. 2022). Slope, slope aspect, and curvature affect water flow, erosion, and fire spread, with steeper slopes either mitigating or accelerating these hazards (Dottori et al. 2018; Trang et al. 2022). The Topographic Wetness Index (TWI) and Stream Power Index (SPI) help quantify water accumulation and erosion risks. Vegetation density, assessed using the Normalized Difference Vegetation Index (NDVI), impacts both flood absorption and fire fuel availability (Abedi Gheshlaghi et al. 2021; Gonzalez-Arqueros et al. 2018). Road and river proximity also influence flood and fire dynamics, while land cover, lithology, and geohydrology influence water retention and fire susceptibility (Ha et

al. 2023; Hosseini and Lim 2022). Rainfall patterns and temperatures, particularly during dry seasons, further contribute to both flood and wildfire risks (Abram et al. 2021; Ahmadlou et al. 2018). These factors are modeled using data from satellite imagery, DEMs, and long-term climate records.

Chapter 3.4 (Experimental Process): All references to landslide susceptibility in this chapter have been eliminated to avoid confusion, as the experimental process is focused solely on wildfires and floods.

**Specific comment 3:** The formula for GINI impurity (Line 218) is wrong: the factor is the sum of pairwise products of the probabilities for each class, thus the correct formula should be:

$$\sum_{i=1}^{J} P_i(1 - P_i) = \sum_{i=1}^{J} P_i - \sum_{i=1}^{J} P_i^2 = 1 - \sum_{i=1}^{J} P_i^2 \tag{1}$$

Moreover, it could be useful for assessing the robustness of the hyperparameter tuning to know which was the range the range of the hyperparameters tested in the Cross Validation and not only the final selection (Table 1), either in the main chapter or in the supplementary materials.

**Response:** Thank you for pointing out the error in the Gini impurity formula and for your suggestion regarding the hyperparameter tuning process. We have corrected the formula in the manuscript to accurately reflect the sum of pairwise products of probabilities for each class. The updated formula now reads:

$$\sum_{i=1}^{J} P_i(1 - P_i) = \sum_{i=1}^{J} P_i - \sum_{i=1}^{J} P_i^2 = 1 - \sum_{i=1}^{J} P_i^2 \tag{1}$$

We added a range of hyperparameters in Table 1 according to your comment. The scikit-optimize's grid search performs iterative assessments using the training data to select the hyperparameter combination that optimizes a chosen performance metric (ROC and AUC) on the testing dataset. The best optimal hyperparameter combinations for each model are determined based on these performance metrics (in the last colum of Table 1).

**Table. 1** The hyperparameter values in the optimization process.

| Model | Optimized Hyperparameter | Explanation | Lower and upper limits | Optimal value |
|---|---|---|---|---|
| CART | max_Nodes | The maximum number of leaf nodes in each tree. | 2-500 | 150 |
| | minLeafPopulation | Only create nodes whose training set contains at least this many points. | 1-10 | 2 |

| | | | | |
|---|---|---|---|---|
| RF | numberOfTrees | The number of decision trees to create. | 100 – 1000 | 200 |
| | minLeafPopulation | Only create nodes whose training set contains at least this many points. | 1-10 | 1 |
| | bagFraction | The fraction of input to bag per tree. | 0.1 – 1.0 | 0.7 |
| | seed | The randomization seed. | 0 - 42 | 23 |

**References**

Abedi Gheshlaghi, H., Feizizadeh, B., Blaschke, T., Lakes, T., and Tajbar, S.: Forest fire susceptibility modeling using hybrid approaches, Transactions in GIS, 25, 311-333, https://doi.org/10.1111/tgis.12688, 2021.

Abram, N. J., Henley, B. J., Sen Gupta, A., Lippmann, T. J. R., Clarke, H., Dowdy, A. J., Sharples, J. J., Nolan, R. H., Zhang, T., Wooster, M. J., Wurtzel, J. B., Meissner, K. J., Pitman, A. J., Ukkola, A. M., Murphy, B. P., Tapper, N. J., and Boer, M. M.: Connections of climate change and variability to large and extreme forest fires in southeast Australia, Communications Earth & Environment, 2, 10.1038/s43247-020-00065-8, 2021.

Ahmadlou, M., Karimi, M., Alizadeh, S., Shirzadi, A., Parvinnejhad, D., Shahabi, H., and Panahi, M.: Flood susceptibility assessment using integration of adaptive network-based fuzzy inference system (ANFIS) and biogeography-based optimization (BBO) and BAT algorithms (BA), Geocarto International, 34, 1252-1272, 10.1080/10106049.2018.1474276, 2018.

Askar, R., Bragança, L., and Gervásio, H.: Adaptability of Buildings: A Critical Review on the Concept Evolution, Applied Sciences, 11, 10.3390/app11104483, 2021.

Bui, Q. D., Luu, C., Mai, S. H., Ha, H. T., Ta, H. T., and Pham, B. T.: Flood risk mapping and analysis using an integrated framework of machine learning models and analytic hierarchy process, Risk Anal, 10.1111/risa.14018, 2022.

Carreño, M. L., Cardona, O. D., and Barbat, A. H.: A disaster risk management performance index, Natural Hazards, 41, 1-20, https://doi.org/10.1007/s11069-006-9008-y, 2007.

Dottori, F., Martina, M. L. V., and Figueiredo, R.: A methodology for flood susceptibility and vulnerability analysis in complex flood scenarios, Journal of Flood Risk Management, 11, S632-S645, https://doi.org/10.1111/jfr3.12234, 2018.

Gonzalez-Arqueros, M. L., Mendoza, M. E., Bocco, G., and Solis Castillo, B.: Flood susceptibility in rural settlements in remote zones: The case of a mountainous basin in the Sierra-Costa region of Michoacan, Mexico, Journal of environmental management, 223, 685-693, https://doi.org/10.1016/j.jenvman.2018.06.075, 2018.

Ha, H., Bui, Q. D., Nguyen, H. D., Pham, B. T., Lai, T. D., and Luu, C.: A practical approach to flood hazard, vulnerability, and risk assessing and mapping for Quang Binh province, Vietnam, Environment, Development and Sustainability, 25, 1101-1130, 10.1007/s10668-021-02041-4, 2023.

Hosseini, M. and Lim, S.: Gene expression programming and data mining methods for bushfire susceptibility mapping in New South Wales, Australia, Natural Hazards, 113, 1349-1365, 10.1007/s11069-022-05350-7, 2022.

Pourtaghi, Z. S., Pourghasemi, H. R., Aretano, R., and Semeraro, T.: Investigation of general indicators influencing on forest fire and its susceptibility modeling using different data mining techniques, Ecological Indicators, 64, 72-84, 10.1016/j.ecolind.2015.12.030, 2016.

Schneiderbauer, S. and Ehrlich, D.: Risk, hazard and people's vulnerability to natural hazards, A review of definitions, concepts and data. European Commission Joint Research Centre. EUR, 21410, 40, 2004.

Trang, P. T., Andrew, M. E., Chu, T., and Enright, N. J.: Forest fire and its key drivers in the tropical forests of northern Vietnam, International Journal of Wildland Fire, 31, 213-229, 10.1071/wf21078, 2022.